# IMPROVING INFORMATION RETENTION IN LARGE SCALE ONLINE CONTINUAL LEARNING

## ABSTRACT

Given a stream of data sampled from non-stationary distributions, online continual learning (OCL) aims to adapt efficiently to new data while retaining existing knowledge. The typical approach to address information retention (the ability to retain previous knowledge) is keeping a replay buffer of a fixed size and computing gradients using a mixture of new data and the replay buffer. Surprisingly, the recent work (Cai et al., 2021) suggests that information retention remains a problem in large scale OCL even when the replay buffer is unlimited, *i.e.*, the gradients are computed using all past data. This paper focuses on this peculiarity to understand and address information retention. To pinpoint the source of this problem, we theoretically show that, given limited computation budgets at each time step, even without strict storage limit, naively applying SGD with constant or constantly decreasing learning rates fails to optimize information retention in the long term. We propose using a moving average family of methods to improve optimization for non-stationary objectives. Specifically, we design an adaptive moving average (AMA) optimizer and a moving-average-based learning rate schedule (MALR). We demonstrate the effectiveness of AMA+MALR on large-scale benchmarks, including Continual Localization (CLOC), Google Landmarks, and ImageNet. Code will be released upon publication.

## 1 INTRODUCTION

Supervised learning commonly assumes that the data is independent and identically distributed (iid.). This assumption is violated in practice when the data comes from a non-stationary distribution that evolves over time. Continual learning aims to solve this problem by designing algorithms that efficiently learn and retain knowledge over time from a data stream. Continual learning can be classified into online and offline. The offline setting (Li & Hoiem, 2017) mainly limits the storage: only a fixed amount of training data can be stored at each time step. The computation is not limited in offline continual learning: the model can be trained from scratch until convergence at each step. In contrast, the online setting only allows a limited amount of storage and computation at each time step.

A number of metrics can be used to evaluate a continual learner. If the model is directly evaluated on the incoming data, the objective is *learning efficacy*; this evaluates the ability to efficiently adapt to new data. If the model is evaluated on historical data, the objective is *information retention*; this evaluates the ability to retain existing knowledge. These two objectives are in conflict, and their trade-off is known as the plasticity-stability dilemma (McCloskey & Cohen, 1989).

Following recent counterintuitive results, we single out information retention in this work. A common assumption in the continual learning literature is that information retention is only a problem due to the storage constraint. Take replay-buffer-based methods as an exemplar. It is tacitly understood that since they cannot store the entire history, they forget past knowledge which is not stored. However, this intuition is challenged by recent empirical results. For example, Cai et al. (2021) show that the information retention problem persists even when past data is stored in its entirety. We argue that, at least in part, the culprit for information loss is optimization.

Direct application of SGD to a continual data stream is problematic. Informally, consider the learning rate (*i.e.*, step size). It needs to decrease to zero over time to guarantee convergence (Ghadimi & Lan, 2013). However, this cannot be applied to a continual and non-iid. stream since infinitesimal learning rates would simply ignore the new information and fail to adapt, resulting in underfitting.

This underfitting would worsen over time as the distribution continues to shift. We formalize this problem and further show that there is no straightforward method to control this trade-off, and this issue holds even when common adaptive learning rate heuristics are applied.

Orthogonal to continual learning, one recently proposed remedy to guarantee SGD convergence with high learning rates is using the moving average of SGD iterates (Mandt et al., 2016; Tarvainen & Valpola, 2017). Informally, SGD with large learning rates bounces around the optimum. Averaging its trajectory dampens the bouncing and tracks the optimum better (Mandt et al., 2016). We apply these ideas to OCL for the first time to improve information retention.

To summarize, we theoretically analyze the behavior of SGD for OCL. Following this analysis, we propose a moving average strategy to optimize information retention. Our method uses SGD with large learning rates to adapt to non-stationarity, and utilizes the average of SGD iterates for better convergence. We propose an adaptive moving average (AMA) algorithm to control the moving average weight over time. Based on the statistics of the SGD and AMA models, we further propose a moving-average-based learning rate schedule (MALR) to better control the learning rate. Experiments on Continual Localization (CLOC) (Cai et al., 2021), Google Landmarks (Weyand et al., 2020), and ImageNet (Deng et al., 2009) demonstrate superior information retention and long-term transfer for large-scale OCL.

## 2 RELATED WORK

**Optimization in OCL.** OCL methods typically focus on improving learning efficacy. Cai et al. (2021) proposed several strategies, including adaptive learning rates, adaptive replay buffer sizes, and small batch sizes. Hu et al. (2020) proposed a new optimizer, ConGrad, which, at each time step, adaptively controls the number of online gradient descent steps (Hazan, 2019) to balance generalization and training loss reduction. Our work instead focuses on information retention and proposes a new optimizer and learning rate schedule that trade off learning efficacy to improve long-term transfer. In terms of replay buffer strategies, mixed replay (Chaudhry et al., 2019), originating from offline continual learning, forms a minibatch by sampling half of the data from the online stream and the other half from the history. It has been applied in OCL to optimize learning efficacy (Cai et al., 2021). Our work uses pure replay instead to optimize information retention, where a minibatch is formed by sampling uniformly from all history.

**Continual learning algorithms.** We focus on the optimization aspect of OCL. Other aspects, such as the data integration (Aljundi et al., 2019b) and the sampling procedure of the replay buffer (Aljundi et al., 2019a; Chrysakis & Moens, 2020), are complementary and orthogonal to our study. These aspects are critical for a successful OCL strategy and can potentially be used in conjuction with our optimizers. Offline continual learning (Li & Hoiem, 2017; Kirkpatrick et al., 2017) aims to improve information retention with limited storage. Unlike the online setting, SGD works in this case since the model can be retrained until convergence at each time step. We refer the readers to Delange et al. (2021) for a detailed survey of offline continual learning algorithms.

**Moving average in optimization.** We propose a new moving-average-based optimizer for OCL. Although we are the first to apply this idea to OCL, moving average optimizers have been widely utilized for convex (Ruppert, 1988; Polyak & Juditsky, 1992) and non-convex optimization (Izmailov et al., 2018; Maddox et al., 2019; He et al., 2020). Beyond supervised learning (Izmailov et al., 2018; Maddox et al., 2019), the moving average model has also been used as a teacher of the SGD model in semi-supervised (Tarvainen & Valpola, 2017) and self-supervised learning (He et al., 2020). The moving average of stochastic gradients (rather than model weights) has also been widely used in ADAM-based optimizers (Kingma & Ba, 2014).

**Continual learning benchmarks.** We need a large-scale and realistic benchmark to evaluate OCL. For language modeling, Hu et al. (2020) created the Firehose benchmark using a large stream of Twitter posts. The task of Firehose is continual per-user tweet prediction, which is self-supervised and multi-task. For visual recognition, Lin et al. (2021) created the CLEAR benchmark by manually labeling images on a subset of YFCC100M (Thomee et al., 2016). Though the images are ordered in time, the number of labeled images is small (33K). Cai et al. (2021) proposed the continual localization (CLOC) benchmark using a subset of YFCC100M with time stamps and geographic locations. The task of CLOC is geolocalization, which is formulated as image classification. CLOC

has significant scale (39 million images taken over more than 8 years) compared to other benchmarks. Due to the large scale and natural distribution shifts, we use CLOC as the main dataset to study OCL.

## 3 PRELIMINARIES

In this section, we formalize the online continual learning problem and introduce our notation. We then discuss the dataset and the metrics we use.

**Problem definition.** Given the input domain $\mathcal{X}$ and the label space $\mathcal{Y}$, online continual learning learns a parametric function $f(\cdot|\boldsymbol{\theta}) : \mathcal{X} \rightarrow \mathcal{Y}$ that maps an input $\mathbf{x} \in \mathcal{X}$ to the corresponding label $\mathbf{y} \in \mathcal{Y}$. At each time step $t \in \{1, 2, ..., \infty\}$ the learner interacts with the environment as follows:

1. The environment samples the data $\{\mathbf{X}_t, \mathbf{Y}_t\} \sim \pi_t$ and reveals the inputs $\mathbf{X}_t$ to the learner.

2. The learner predicts the label for each datum $\mathbf{x}_t^{i_t} \in \mathbf{X}_t$ via $\hat{\mathbf{y}}_t^{i_t} = f(\mathbf{x}_t^{i_t}|\boldsymbol{\theta}_{t-1})$.

3. The environment reveals the true labels $\mathbf{Y}_t$ and the learner integrates $\{\mathbf{X}_t, \mathbf{Y}_t\}$ into the data pool $\mathbf{S}_t$ via $\mathbf{S}_t = \text{update}(\mathbf{S}_{t-1}, \{\mathbf{X}_t, \mathbf{Y}_t\})$.

4. The learner updates the model $\boldsymbol{\theta}_t$ using $\mathbf{S}_t$.

**Dataset.** Due to the large scale, the smooth data stream, and natural distribution shifts, we mainly use the Continual Localization (CLOC) benchmark (Cai et al., 2021) to study OCL. CLOC formulates image geolocalization as a classification problem. It contains 39 million labeled images for training, 39 thousand images for performance evaluation, and another 2 million images for initial hyperparameter tuning. The images are ordered according to their time stamps, so that images taken earlier are seen by the model first. Images for training and performance evaluation cover the same time span and are sampled uniformly over time. Images for hyperparameter tuning are from a different time span that does not overlap with the training and evaluation data. We refer to the 39 million training images as the *training data*, the 39 thousand evaluation images as the *evaluation data*, and the 2 million images for initial hyperparameter tuning as the *preprocessing data*.

**Evaluation protocol.** We evaluate OCL algorithms using the following metrics.

*Learning Efficacy:* For online learning applications, the model $\boldsymbol{\theta}_t$ is deployed for inference on data from the next time step, i.e., $\mathbf{X}_{t+1}$. In this case, we measure the ability to adapt to new data as

$$P_{LE}(t) = \frac{1}{t} \sum_{j=1}^{t} \text{acc}(\{\mathbf{X}_{j+1}, \mathbf{Y}_{j+1}\}, \boldsymbol{\theta}_j), \tag{1}$$

where $\text{acc}(\mathbf{S}, \boldsymbol{\theta})$ is the average accuracy of the model $\boldsymbol{\theta}$ on a set of data $\mathbf{S}$.

*Information Retention:* When the data for inference comes from the history, we measure the ability to retain previous knowledge as

$$P_{IR}(t) = \text{acc}\Big(\Big\{ \bigcup_{j=1}^{t} \mathbf{X}_j^E, \bigcup_{j=1}^{t} \mathbf{Y}_j^E \Big\}, \boldsymbol{\theta}_t\Big), \tag{2}$$

where $\{\mathbf{X}_j^E, \mathbf{Y}_j^E\}$ is the evaluation data from time step $j$, which has the same distribution as $\{\mathbf{X}_j, \mathbf{Y}_j\}$ but unseen by the model. $P_{IR}(t)$ measures the generalization of $\boldsymbol{\theta}_t$ to historical data.

*Forward Transfer:* When long-term predictions are required, we measure ability to generalize from current time ($t$) to future time steps ($t + k_1$ to $t + k_2$, $k_2 > k_1 \geq 1$) as

$$P_{FT}(t) = \text{acc}\Big(\Big\{ \bigcup_{j=t+k_1}^{t+k_2} \mathbf{X}_j^E, \bigcup_{j=t+k_1}^{t+k_2} \mathbf{Y}_j^E \Big\}, \boldsymbol{\theta}_t\Big). \tag{3}$$

## 4 METHOD

We consider replay-based methods, where the common choice for continual learning is mixed replay (Chaudhry et al., 2019). The objective of mixed replay at time step $t$ is

$$\underset{\boldsymbol{\theta} \in \mathbb{R}^d}{\text{minimize}} \quad l(\{\mathbf{X}_t, \mathbf{Y}_t\}, \boldsymbol{\theta}) + l(\mathbf{S}_{t-1}, \boldsymbol{\theta}), \tag{4}$$

where $\{\mathbf{X}_t, \mathbf{Y}_t\}$ is the data received at time $t$, $\mathbf{S}_{t-1}$ is the historical data accumulated in the replay buffer, and $l(\cdot, \boldsymbol{\theta})$ is the loss function. Mixed replay constructs a minibatch of training data by sampling half from the current time step $t$ and another half from the history (from $t - B_t$ to $t - 1$ where $B_t$ decides the time range). In offline continual learning (Chaudhry et al., 2019), $B_t$ is set to $t - 1$ for full coverage. Cai et al. (2021) adaptively choose $B_t$ to optimize learning efficacy in OCL.

In this paper, we focus on information retention, more specifically its optimization. A reasonable objective for pure information retention is *pure replay*, which is defined as follows at each time step $t$:

$$\underset{\boldsymbol{\theta} \in \mathbb{R}^d}{\text{minimize}} \quad l(\mathbf{S}_t, \boldsymbol{\theta}). \tag{5}$$

Pure replay constructs a minibatch of training data by sampling uniformly from all historical time steps, i.e., from 1 to $t$, which encourages remembering all historical knowledge. As shown in Appx. C.6, optimizing (5) (as opposed to (4)) improves both information retention and forward transfer by sacrificing learning efficacy.

### 4.1 OPTIMIZING INFORMATION RETENTION

When the distribution of $\mathbf{S}_t$ remains unchanged over time (i.e., the loss function is stationary), the standard optimizer for problem (5) is Stochastic Gradient Descent (SGD). This choice is typically carried over to continual learning. At each update iteration $k$ ($k \neq t$ if multiple updates are allowed at each time step of continual learning), SGD updates the model via

$$\boldsymbol{\theta}_k \leftarrow \boldsymbol{\theta}_{k-1} - \alpha_k g_k(\boldsymbol{\theta}_{k-1}), \tag{6}$$

where $\alpha_k$ is the learning rate and $g_k(\boldsymbol{\theta}_{k-1})$ is the stochastic gradient of the objective at iteration $k$.

Although the convergence behavior of SGD is well understood for the case of stationary losses, its implications for OCL are not clear. Hence, we extend the analysis of SGD (Ghadimi & Lan, 2013) to the continual learning case. Denote the loss function at iteration $k$ by $l_k(\boldsymbol{\theta})$ and assume:

**A1** $\|\triangledown l_k(\boldsymbol{\theta}) - \triangledown l_k(\boldsymbol{\theta}')\| \leq L \|\boldsymbol{\theta} - \boldsymbol{\theta}'\|, \forall \boldsymbol{\theta}, \boldsymbol{\theta}' \in \mathbb{R}^d$ and $k$ (Lipschitz smoothness).

**A2** $\mathbb{E}[g_k(\boldsymbol{\theta})] = \triangledown l_k(\boldsymbol{\theta}), \forall \boldsymbol{\theta} \in \mathbb{R}^d$ and $k$ (Unbiased gradient estimates).

**A3** $\|\triangledown l_k(\boldsymbol{\theta}) - g_k(\boldsymbol{\theta})\|^2 \leq \rho^2, \forall \boldsymbol{\theta} \in \mathbb{R}^d$ (Bounded gradient noise).

**A4** $|l_{k+1}(\boldsymbol{\theta}) - l_k(\boldsymbol{\theta})| \leq \chi_k, \forall \boldsymbol{\theta}$ and $k$ (Bounded non-stationarity).

A1 to A3 extend the standard assumptions for SGD analysis (Ghadimi & Lan, 2013) from the stationary case to the non-stationary case. And A4 bounds the degree of non-stationarity between consecutive iterations. The expectations are taken over the randomness of the stochastic gradient noise. With these assumptions, we state the following theorem and defer its proof to Appx. A.2 (also see Appx. A.2 for details about how to interpret this result under limited storage):

**Theorem 1.** *Assume A1 to A4, and let* $\alpha_k < \frac{L}{2}$,

$$\min_{j \in \{0, 1, \ldots, k\}} \mathbb{E}[\|\triangledown l_{j+1}(\boldsymbol{\theta}_j)\|^2] \leq T_1 + T_2 + T_3, \tag{7}$$

*where* $T_1 = \frac{2(l_1(\boldsymbol{\theta}_0) - \mathbb{E}[l_{k+2}(\boldsymbol{\theta}_{k+1})])}{\sum_{j=0}^k (2\alpha_{j+1} - L\alpha_{j+1}^2)}$, $T_2 = \frac{L\rho^2 \sum_{j=0}^k \alpha_{j+1}^2}{\sum_{j=0}^k (2\alpha_{j+1} - L\alpha_{j+1}^2)}$, *and* $T_3 = \frac{2 \sum_{j=0}^k \chi_{j+1}}{\sum_{j=0}^k (2\alpha_{j+1} - L\alpha_{j+1}^2)}$.

Therorem 1 bounds the minimum gradient norm achievable by SGD during OCL. A similar bound in the stationary case (Theorem 2 in Appx. A.1) has only the terms $T_1$ and $T_2$. Hence, $T_3$ is the cost of the non-stationarity. In the stationary case (when $T_3 = 0$), we can simply find the optimal learning rate by trading off between $T_1$ and $T_2$. Specifically, for a constant learning rate, $T_1$ converges to 0 at a linear rate, but $T_2$ is constant. Hence, the strategy for convergence is reducing learning rates to control $T_2$ simultaneously with $T_1$. This result supports the effectiveness of the standard "*reduce-when-plateau*" (RWP) learning rate heuristic in the stationary case. Since the access to the true gradient norm is not practical, RWP often uses the validation accuracy/loss as a surrogate to control the learning rate $\alpha_k$, where we reduce $\alpha_k$ by a certain factor $\beta$ once the validation accuracy/loss plateaus (Goodfellow et al., 2016; He et al., 2016).

In the continual learning setting (when $T_3 \neq 0$), RWP can be less effective. For example, if $\chi_k \geq \chi > 0$ for a constant $\chi$, i.e., the loss function $l_k(\cdot)$ changes at least at a linear rate, then $T_3$ can grow linearly when the learning rate is reduced to 0. Hence in OCL, there can be cases where

simultaneous convergence of $T_1, T_2$, and $T_3$ are not possible. Controlling $T_3$ requires a reasonably large learning rate, whereas using a large learning rate increases $T_2$. Finally, the $\chi_k$ term can cause the learning rate annealing signal in RWP, i.e., the gradient norm or the validation loss/accuracy, to be unstable in OCL. We argue that these are the core obstacles in application of SGD to OCL.

In summary, the major issues hindering the performance of OCL when optimized with SGD are:

**P1** OCL needs large learning rates due to non-stationarity. However, convergence requires smaller learning rates.

**P2** RWP allows the learning rate to be infinitesimal, which prevents the model from adapting to new data.

**P3** The validation performance is not always indicative of learning. E.g., due to the distribution shift of OCL, the validation accuracy can decrease, even when the model is still improving.

To address **P1**, we propose an Adaptive Moving Average (AMA) optimizer in Sec. 4.2. Then, to address **P2** and **P3**, we introduce a moving-average-based learning rate (MALR) schedule in Sec. 4.3.

## 4.2 ADAPTIVE MOVING AVERAGE

Our Adaptive Moving Average (AMA) optimizer extends Exponential Moving Average (EMA) (Tarvainen & Valpola, 2017). Moving average optimizers maintain two models: the SGD model $\boldsymbol{\theta}_k$ and the MA model $\boldsymbol{\theta}_k^{MA}$. At each iteration $k$, $\boldsymbol{\theta}_k$ is first updated with (6). Then, $\boldsymbol{\theta}_k^{MA}$ is updated via

$$\boldsymbol{\theta}_k^{MA} \leftarrow \gamma_k \boldsymbol{\theta}_{k-1}^{MA} + (1 - \gamma_k) \boldsymbol{\theta}_k, \tag{8}$$

where the MA weight $\gamma_k \in [0, 1]$ is a hyperparameter. In EMA, $\gamma_k$ is a constant. In AMA, we adapt $\gamma_k$ over time using population-based search (Jaderberg et al., 2017) to improve the performance.

**Why does AMA help information retention?** Given stationary losses, SGD can converge to local optima by reducing the learning rate. In this case, the final solution found by AMA will not be very different from SGD as we validate in Appx. C.10. In the non-stationary case, SGD needs reasonably large learning rates to track the shifting optimum over time. However, the variance ($T_2$ in (7)) of SGD increases with large learning rates, hurting convergence.

The advantage of AMA is to reduce the variance of SGD. Given stationary losses, Mandt et al. (2016) have shown that SGD with a constant learning rate will eventually bounce around a region centered at the optimum. Averaging the SGD trajectory with uniform weights estimates the region center, i.e., the optimal solution. This observation inspires the design of AMA. Informally, SGD with large learning rates would follow and bounce around the *shifting* optimum in OCL. The moving average of these high-variance iterates tracks the shifting optimum better. As a further informal justification, consider the unfolded AMA update rule (8):

$$\boldsymbol{\theta}_k^{MA} = (1 - \gamma_k) \boldsymbol{\theta}_k + \sum_{i=1}^{k-1} (1 - \gamma_i) \left( \prod_{j=i+1}^{k} \gamma_j \right) \boldsymbol{\theta}_i + \left( \prod_{j=1}^{k} \gamma_j \right) \boldsymbol{\theta}_0, \tag{9}$$

where $\theta_0$ is the initial solution of SGD and AMA. (9) shows that the AMA model $\boldsymbol{\theta}_k^{MA}$ is a linear combination of the SGD trajectory $\{\boldsymbol{\theta}_0, ..., \boldsymbol{\theta}_k\}$. Since $\gamma_k \in [0, 1]$, $\boldsymbol{\theta}_k^{MA}$ lies inside the convex hull of $\{\boldsymbol{\theta}_0, ..., \boldsymbol{\theta}_k\}$. The population-based search aims to find the best point within the convex hull by adapting $\gamma_k$.

We summarize our method in Alg. 1. To adapt $\gamma_k$ over time, we maintain two MA models, $\boldsymbol{\theta}^{MA_1}$ and $\boldsymbol{\theta}^{MA_2}$, and update them using different weights, $\gamma^{MA_1}$ and $\gamma^{MA_2}$ (Line 4). To decide the best model and weight, we use the validation accuracy $Acc_1$ and $Acc_2$ (Line 5), which we define next. Once every $K_W$ iterations, we copy the parameter of the best MA model to the other one, and adapt the MA weights (Line 7 to 10). In all our experiments, we set the initial MA weight $\gamma_0$ to 0.99, the MA weight adjustment coefficient $\delta$ to 5, and the interval $K_W$ to adjust weights to 10000 iterations.

We continuously track a validation metric for adaptation. At each time step $t$, a holdout *information retention validation set* $\mathbf{S}_t^V$ is updated, which has the same distribution as $\mathbf{S}_t$. To maintain $\mathbf{S}_t^V$ online, whenever new data $\{\mathbf{X}_t, \mathbf{Y}_t\}$ is received, we put a random subset (5% in all experiments) into $\mathbf{S}_t^V$ and put the rest into $\mathbf{S}_t$. We compute the *information retention validation accuracy/loss* on $\mathbf{S}_t^V$ in an online fashion (described in Appx. B). In this way, we can parallelize validation with model updates,

---

**Algorithm 1** Adaptive Moving Average (AMA)

---

**Require:** Initial model $\theta_0$, initial moving average weight $\gamma_0$, MA weight adjustment coefficient $\delta(>0)$, MA weight update interval $K_W$.
1: $\theta^{MA_1} \leftarrow \theta_0, \theta^{MA_2} \leftarrow \theta_0, \gamma^{MA_1} \leftarrow \gamma_0, \gamma^{MA_2} \leftarrow \frac{\gamma_0}{\delta}, Acc_1 \leftarrow 0, Acc_2 \leftarrow 0, i_{best} \leftarrow 1$
2: **for** $k \in \{1, 2, 3, ..., \infty\}$ **do**
3:     Update the SGD model $\theta_k$
4:     Update $\theta^{MA_1}$ using $\gamma^{MA_1}$ and $\theta^{MA_2}$ using $\gamma^{MA_2}$
5:     Update validation accuracy $Acc_1$ and $Acc_2$ online
6:     **If** $Acc_1 > Acc_2$ **then** $i_{best} = 1$ **else** $i_{best} = 2$
7:     **if** $k \mod K_W = 0$ **then**
8:         Copy the weights of $\theta^{MA_{i_{best}}}$ to the other. $Acc_1 \leftarrow 0, Acc_2 \leftarrow 0$.
9:         Increase $\gamma^{MA_1} \& \gamma^{MA_2}$ by a factor of $\delta$ if $i_{best} = 1$, and decrease otherwise.
10:     **end if**
11: **end for**

---

and choose the best MA model for inference anytime during OCL. We use the information retention validation accuracy/loss both in Algorithm 1 and later for learning rate control.

We propose several strategies to reduce the overhead of MA and population-based search. The overhead from MA lies in 1) updating the MA model and 2) recomputing the batch normalization (BN) statistics (Ioffe & Szegedy, 2015). To solve 1, we perform the MA update only every $K_M$ iterations. We set $K_M = 10$ in all experiments and do not observe a performance drop. To solve 2, we replace BN with group normalization (GN) + weight standardization (WS) (Qiao et al., 2019), which does not store normalization statistics. We evaluate its impact in Appx. C.1. The overhead from population-based search lies in the online validation operation (Line 5). Each operation requires one forward pass on a batch of validation data. To reduce the overhead, we execute Line 5 every $K_V = 20$ iterations. See Appx. B for a detailed version of Algorithm 1 with the above strategies.

### 4.3 Moving-Average-based Learning Rate Schedule (MALR)

As discussed in (**P3**), monitoring only the validation accuracy is not sufficient for learning rate control as the effects of learning and distribution shifts counteract each other. Interestingly, AMA provides a better signal for learning rate control as a by-product. Specifically, we use the difference between *information retention validation loss/accuracy* of the MA model $\theta^{MA_{i_{best}}}$ and the SGD model $\theta_k$. We denote this difference at iteration $k$ by $\sigma_k$. Intuitively, since AMA performs better than SGD given non-minimal learning rates, $\sigma_k$ can be used to estimate the performance gain obtainable by learning rate reduction. To control the learning rate reduction speed, we do not reduce learning rates unless $\sigma_k$ plateaus, indicating that we cannot achieve better long-term performance gain. On the other hand, when the learning rate is too small, potentially incapable of adapting to new data, $\sigma_k$ will also be very small. Hence, we do not reduce the learning rate unless $\sigma_k$ exceeds a certain threshold.

In summary, MALR reduces the learning rate when the following conditions are satisfied. **C1**: The information retention validation performance (of the MA model) does not improve for $K_R$ iterations. (This is the original RWP condition.) **C2**: $\sigma_k$ does not increase for $K_R$ iterations. **C3**: $\sigma_k > \epsilon$.

**Computational complexity.** The average computational complexity of SGD+RWP per iteration is $c_{\text{SGD+RWP}} = (\frac{K_V+1}{K_V})c_F + c_G + c_U$, where $c_F$ is the complexity of one forward pass, $K_V$ is the interval to do one online validation step, $c_G$ and $c_U$ are respectively the complexity for computing the gradient and updating the model. The complexity of AMA+MALR is $c_{\text{AMA+MALR}} = (\frac{K_V+3}{K_V})c_F + c_G + (\frac{K_M+2}{K_M})c_U$, where $K_M$ is the interval to do one MA update step. When $K_V$ and $K_M$ are large, the overhead of AMA+MALR is small.

## 5 Experiments

**Implementation details.** We mainly use CLOC (Cai et al., 2021) to study OCL at large scale. Similar to the setting of Cai et al. (2021), the model receives 256 images at each time step. We use the ResNet-50 architecture, a batch size of 256 and tune the initial hyperparameters on the preprocessing data. We set the initial learning rate $\alpha_0$ to 0.025, the momentum of SGD to 0.9, the weight decay to

(a) Information reten- (b) Log-scaled learn- (c) Validation loss (d) Validation loss (e) $\sigma_k$ over time.
tion over time. ($\uparrow$) ing rate over time. over time. ($\downarrow$) (zoomed view). ($\downarrow$)

Figure 1: **Results on CLOC**. (a): AMA+MALR outperformed all competitors. Among optimizers, AMA outperformed SGD in both RWP and MALR. (b): MALR effectively annealed the learning rate over time, outperforming RWP and CLR. (c) & (d): Under distribution shifts, the validation loss increased for constant learning rates in both long and short terms. This made RWP reduce the learning rate too fast. (e): Given a constant learning rate, $\sigma_k$ in MALR first increased and then plateaued. $\sigma_k$ declined with the learning rate reduction. This trend was stable even under distribution shifts.

$1e-4$, and the loss to cross-entropy. For RWP, we reduce the learning rate by half if the information retention validation loss stops decreasing for $K_R = 60000$ training iterations. For MALR, we set $\sigma_k$ as the difference of the information retention validation accuracy between the MA and the SGD models, and set $\epsilon$ in **C3** to $3.0\%$. To decouple the effect of the storage limit and better focus on studying the problem of limited computation in large scale OCL, we use a replay buffer size of 40 million images in main experiments. Appx. **C.4** ablates the effect of replay buffer sizes and shows that large scale OCL only requires a reasonably large replay buffer (4 million images for CLOC). We use the same set of hyperparameters introduced by AMA/MALR in all experiments to verify their robustness. (See Appx. **C.11** for further details.) All models are trained with 8 Nvidia Geforce Rtx 2080 Ti GPUs. We refer to the information retention validation loss/accuracy throughout the experiments as the validation loss/accuracy. The arrows ($\uparrow$/$\downarrow$) in the caption of each figure point towards the direction of improvement.

## 5.1 MAIN RESULT: EFFECTIVENESS OF AMA AND MALR

To study the effectiveness of AMA + MALR, we compare it against SGD + RWP (SGD is used as the base optimizer due to high performance, see Appx. **C.5** for similar experiments on ADAM). To separate the effects of optimizers and learning rate schedules, we also compare against AMA + RWP, AMA + constant learning rate (CLR), the SGD model in AMA+MALR, and SGD+CLR (using training + validation data for training since no online hyperparameter adaptation is needed). We allow 80 training iterations per time step, so that the number of training iterations is large enough and learning rate annealing is necessary. We evaluate information retention using $P_{IR}(t)$.

As shown in Fig. 1a, AMA+MALR outperformed all competitors. The "jumps" of the performance curves were due to the learning rate change in RWP or MALR. Note that the information retention in Fig. 1a was not monotonically increasing because of the distribution shifts. In terms of the optimizer, AMA was better than SGD in all learning rate schedules. Moreover, the performance gap was especially large for large learning rates, validating **P1**. In terms of the learning rate schedule, MALR outperformed both RWP and CLR. Specifically, the distribution shift made the validation loss increase over time for constant learning rates (Fig. 1c and 1d). This phenomenon caused RWP to reduce the learning rate too fast (Fig. 1b). As a result, AMA+RWP and SGD+RWP performed even worse than SGD+CLR in the long term. In contrast, MALR effectively controlled the learning rate reduction speed and the minimum learning rate with the help of the new signal $\sigma_k$ (Fig. 1e), which was more robust to distribution shifts than the validation loss/accuracy.

## 5.2 ABLATIONS

MALR extends RWP by introducing **C2** and **C3**. To understand the effect of each condition, we compare AMA+MALR to versions where **C2** is removed (AMA+MALR+No C2), **C3** is removed (AMA+MALR+No C3), and both are removed (AMA+RWP).

As shown in Fig. 2a, AMA+MALR+No C3 failed to limit the minimum learning rate (**P2**), and performed worse than AMA+MALR in the long term. AMA+MALR+No C2 reduced the learning rate too quickly (**P3**) at the early stage compared to AMA+MALR. With both **C2** and **C3** removed, AMA+RWP performed the worst in all methods. As shown in Fig. 2c, the performance of all competitors dropped when $\sigma_k$ was too low, which happened when the learning rate was too small

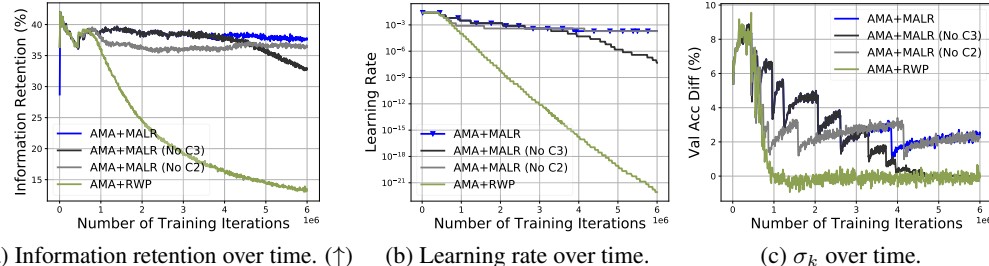

Figure 2: **Ablation for MALR.** AMA+MALR outperformed all competitors. AMA+MALR+No C2 reduced the learning rate too fast at early stages, causing **P3**. AMA+MALR+No C3 did not limit the minimum learning rate, causing **P2**. AMA+RWP performed worst.

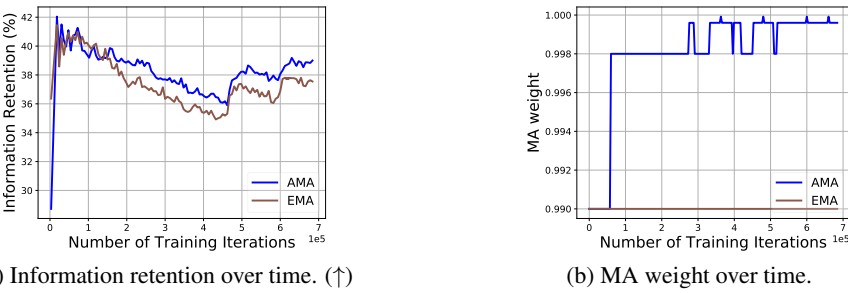

Figure 3: **AMA vs EMA**. AMA performed better than EMA by adapting the MA weight over time.

(Fig. 2b). Hence, $\sigma_k$ is an effective learning rate annealing signal for OCL. Appx. C.2 further analyzes the effect of using suboptimal learning rates. To summarize, large learning rates led to poor convergence and hurt performance on data *from all time ranges*. On the other hand, small learning rates prevented the model from adapting to new data.

**AMA vs EMA.** We study the effect of population-based MA weight adaptation. We compare AMA with EMA and use the learning rate generated by AMA+MALR on both models to remove the effect of learning rate annealing. Due to resource limitations, we train EMA only for 700 thousand iterations. As shown in Fig. 3, AMA outperformed EMA by adapting the MA weight over time.

**Further analyses in the appendix.** Due to the space limit, we provide the following analyses in the appendix. *Effect of batch sizes in Appx. C.3*: Heavy parallelization by increasing batch sizes and decreasing the number of training iterations has negative impact on all OCL metrics. *Effect of training objectives, i.e.,* (5) *vs* (4)*, in Appx. C.6*: Optimizing information retention (using (5)) is more suitable for both information retention and long-term forward transfer, at the cost of sacrificing learning efficacy. More importantly, increasing the computation budget for optimizing information retention improved all OCL metrics, whereas increasing the computation budget for optimizing learning efficacy (using (4)) *hurt* information retention and forward transfer.

### 5.2.1 EFFECTIVENESS OF AMA BEYOND CLOC

**Google Landmarks.** We apply AMA+MALR to *Google Landmarks v2* (Weyand et al., 2020), which contains images from different landmarks (each representing a class). We construct our continual dataset using a subset of 580 thousand training images with time stamps. We refer to the constructed dataset as *Continual Google LandMarks (CGLM)*. Similar to CLOC, 256 new images are revealed at each time step, and 80 training iterations are allowed per time step (see Appx. C.7 for more implementation details). As shown in Fig. 4-a, AMA performed better than SGD in CGLM. Since CGLM is small-scale, the learning rate of both RWP and MALR remained constant throughout most of the training. Hence, we only reported the performance of AMA+MALR and the SGD model. To further check this limitation, we also looked at the first 580K images of CLOC in Figs. 4-c and 4-d. Similar to CGLM, the relative merit of different learning rate schedules was unclear. Hence, a large-scale dataset is necessary for analyzing OCL learning rate schedules.

**ImageNet.** We test our method on task-based benchmarks synthesized from ImageNet. Specifically, we create a 100-task continual learning problem by randomly dividing ImageNet classes, so that each task contains data from 10 classes. During training, we allow $\frac{120|\mathbf{X}_t|}{256}$ training iterations per task,

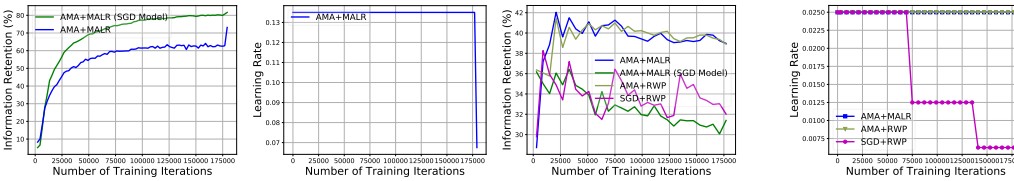

(a) Information retention on CGLM (↑).

(b) Learning rate over time on CGLM.

(c) Information retention on the first 580K CLOC images.

(d) Learning rate for the first 580K images of CLOC.

Figure 4: **Results on the CGLM**. AMA performed better than SGD. Given only 580K images, the difference of learning rate schedules are unclear for CGLM ((a) & (b)) and CLOC ((c) & (d)).

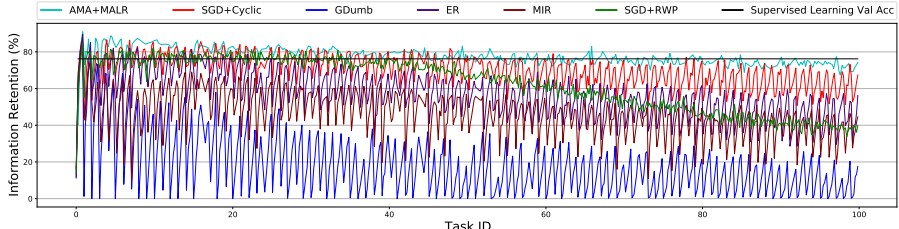

Figure 5: **ImageNet results (↑)**. AMA+MALR out-performed both SGD+RWP and SGD+Cyclic. The final validation accuracy is close to the supervised learning model. Due to the implicit computation increase in each model update iteration, OCL methods performed worse than simple SGD+Cyclic when the memory is sufficiently large and the computation is explicitly limited.

where $|\mathbf{X}_t|$ is the number of images of task $t$ and 256 is the training batch size. In this setting, the total number of SGD steps of a continual learner trained on all tasks is the same as the standard supervised learning model trained for 120 epochs. During evaluation, we compute the average accuracy on validation data from all revealed tasks. To demonstrate the online performance, we perform multiple evaluations during each task. Since clear task boundaries are available, we also compare against SGD+Cyclic, where the cosine learning rate schedule is used in each task. We further compare against methods designed for handling storage limitations: GDumb (Prabhu et al., 2020), ER (Aljundi et al., 2018), and MIR (Aljundi et al., 2019a). We explicitly limit the computation budget of ER, GDumb, and MIR to be similar to other methods. See Appx. C.8 for more details about the experimental setup and hyperparameter settings.

As shown in Fig. 5, AMA+MALR outperformed both SGD+RWP and SGD+Cyclic. Comparing to SGD+Cyclic, AMA+MALR had better intermediate and final performance during each task, making it more suitable for online applications. Using the same number of SGD steps, the final validation accuracy of AMA+MALR was close to supervised learning. On the other hand, ER, MIR, and GDumb performed worse than SGD+Cyclic since implicit computation increase in each model update step made them require more computation/time to converge. Note that here we allow the replay buffer to store all past data. This result demonstrates that limited computation and limited storage are orthogonal problems in OCL. The are both important and require dedicated treatments. See Appx. C.9 for similar results with limited storage.

## 6 CONCLUSION

We studied the problem of improving information retention in online continual learning (OCL). We proposed to use pure replay as the objective for optimizing information retention. We provided theoretical analysis of the convergence of SGD in OCL. We also proposed an Adaptive Moving Average (AMA) Optimizer and a Moving-Average-based Learning Rate Schedule (MALR) to optimize the pure replay objective online. Experiments on several large-scale continual learning benchmarks demonstrate the effectiveness of AMA+MALR. In terms of limitations and future work, we used two MA models to adapt the MA weights, which increases the GPU memory footprint to store the network weights for MA models (but not the gradient information as in the SGD model). Designing more memory-efficient techniques to adapt MA weights is an interesting future research direction. Furthermore, this work mainly focused on studying the problem of limited computation in OCL. Developing effective solutions to address the computation and storage aspects together is an interesting and important future direction.

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

## A    CONVERGENCE ANALYSIS OF SGD

This section analyzes the convergence of SGD in both supervised learning (SL) and online continual learning (OCL). The purpose of the analysis is to demonstrate the difference between the optimization problem in SL and OCL, and explain intuitively why SGD+RWP works in SL but can fail in OCL.

### A.1 CONVERGENCE ANALYSIS IN SUPERVISED LEARNING

In supervised learning, we are given a fixed set of training data, and we solve the following stationary optimization problem

$$\underset{\boldsymbol{\theta} \in \mathbb{R}^d}{\text{minimize}} \quad l(\boldsymbol{\theta}). \tag{10}$$

In each iteration k, SGD updates the model $\boldsymbol{\theta}_k$ using the stochastic gradient, from a batch of training data, via,

$$\boldsymbol{\theta}_k \leftarrow \boldsymbol{\theta}_{k-1} - \alpha_k g(\boldsymbol{\theta}_{k-1} | \mathcal{E}_k), \tag{11}$$

where $\mathcal{E}_k$ denotes the random variable that determines the noise of the stochastic gradient.

The standard assumptions (Ghadimi & Lan, 2013) for analyzing the non-convex case of supervised learning is

**A5** Lipschitz Smoothness with constant $L \geq 0$ : $\|\nabla l(\boldsymbol{\theta}) - \nabla l(\boldsymbol{\theta}')\| \leq L \|\boldsymbol{\theta} - \boldsymbol{\theta}'\|, \forall \boldsymbol{\theta}, \boldsymbol{\theta}' \in \mathbb{R}^d$, where $\|\cdot\|$ is the Euclidean Norm.

**A6** Unbiased gradient estimation: $\mathbb{E}_{\mathcal{E}}[g(\boldsymbol{\theta}|\mathcal{E})] = \nabla l(\boldsymbol{\theta}), \forall \boldsymbol{\theta} \in \mathbb{R}^d$.

**A7** Bounded gradient noise with constant $\rho \geq 0$: $\|\nabla l(\boldsymbol{\theta}) - g(\boldsymbol{\theta}|\mathcal{E})\|^2 \leq \rho^2, \forall \boldsymbol{\theta} \in \mathbb{R}^d$.

Under these assumptions, we re-produce the standard convergence result of SGD in the non-convex SL case:

**Theorem 2.** *Assume A5 to A7, and let $\alpha_k < \frac{L}{2}$,*

$$\min_{j \in \{0,...,k\}} \mathbb{E}_{\mathcal{E}_1,...,\mathcal{E}_j}[\|\nabla l(\boldsymbol{\theta}_j)\|^2] \leq \frac{2(l(\boldsymbol{\theta}_0) - \mathbb{E}_{\mathcal{E}_1,...,\mathcal{E}_{k+1}}[l(\boldsymbol{\theta}_{k+1})])}{\sum_{j=0}^k (2\alpha_{j+1} - L\alpha_{j+1}^2)} + \frac{L\rho^2 \sum_{j=0}^k \alpha_{j+1}^2}{\sum_{j=0}^k (2\alpha_{j+1} - L\alpha_{j+1}^2)}. \tag{12}$$

*Proof.* Following the proof of Theorem 2.1 until (2.11) in (Ghadimi & Lan, 2013), we have

$$\sum_{j=0}^k (2\alpha_{j+1} - L\alpha_{j+1}^2)\mathbb{E}_{\mathcal{E}_1,...,\mathcal{E}_j}[\|\nabla l(\boldsymbol{\theta}_j)\|^2] \leq 2(l(\boldsymbol{\theta}_0) - \mathbb{E}_{\mathcal{E}_1,...,\mathcal{E}_{k+1}}[l(\boldsymbol{\theta}_{k+1})]) + L\rho^2 \sum_{j=0}^k \alpha_{j+1}^2. \tag{13}$$

And since

$$(\sum_{j=0}^k (2\alpha_{j+1} - L\alpha_{j+1}^2)) \min_{l \in \{0,...,k\}} \mathbb{E}_{\mathcal{E}_1,...,\mathcal{E}_l}[\|\nabla l(\boldsymbol{\theta}_l)\|^2] \leq \sum_{j=0}^k (2\alpha_{j+1} - L\alpha_{j+1}^2)\mathbb{E}_{\mathcal{E}_1,...,\mathcal{E}_j}[\|\nabla l(\boldsymbol{\theta}_j)\|^2], \tag{14}$$

we immediately have (12) by combining (13) with (14) and dividing both sides by $\sum_{j=0}^k (2\alpha_{j+1} - L\alpha_{j+1}^2)$. □

Theorem 2 tells us that the convergence of SGD can be guaranteed by properly setting the learning rate $\alpha_k$. Denote $T_4 = \frac{2(l(\boldsymbol{\theta}_0) - \mathbb{E}_{\mathcal{E}_1,...,\mathcal{E}_{k+1}}[l(\boldsymbol{\theta}_{k+1})])}{\sum_{j=0}^k (2\alpha_{j+1} - L\alpha_{j+1}^2)}$ and $T_5 = \frac{L\rho^2 \sum_{j=0}^k \alpha_{j+1}^2}{\sum_{j=0}^k (2\alpha_{j+1} - L\alpha_{j+1}^2)}$ as the first and the second terms at the right hand side of (12), which are of the same form as $T_1$ and $T_2$ of (7). To ensure that $T_4$ goes to 0, we need to make sure that

$$\lim_{k \to \infty} \sum_{j=0}^k (2\alpha_{j+1} - L\alpha_{j+1}^2) = \infty. \tag{15}$$

And to ensure that $T_5$ goes to 0, we need to ensure that

$$\lim_{k \to \infty} \frac{\sum_{j=0}^k \alpha_{j+1}^2}{\sum_{j=0}^k (2\alpha_{j+1} - L\alpha_{j+1}^2)} = 0. \tag{16}$$

For example, one can set $\alpha_k = \mathcal{O}(\sqrt{k})$ so that both (15) and (16) are satisfied. Theorem 2 also provides insights of why RWP is effective in SL. Given a constant learning rate $\alpha_k = \alpha$, we can see that $T_4 = \mathcal{O}(\frac{1}{k})$ after $k$ iterations, i.e., it converges with linear speed. However, the second term $T_5 = \frac{L\rho^2\alpha}{2-L\alpha}$ is a constant over time. Hence, the gradient norm will first decrease and then converge to a constant given a constant learning rate. And the plateau of the gradient norm is a signal that we need to reduce the learning rate to further improve the model. In practice, the validation loss/accuracy is often used as a surrogate of the gradient norm.

## A.2 CONVERGENCE ANALYSIS IN OCL

We provide a proof of Theorem 1 in this section. Before the proof, we first re-state the problem, the assumptions and the theory in a more detailed format.

At each iteration $k$, we are given a non-stationary objective

$$\underset{\boldsymbol{\theta}}{\text{minimize}} \quad l_k(\boldsymbol{\theta}). \tag{17}$$

In each iteration $k$, SGD updates the model $\boldsymbol{\theta}_k$ using the stochastic gradient $g_k(\boldsymbol{\theta}_{k-1}|\mathcal{E}_k)$ on $l_k(\boldsymbol{\theta}_{k-1})$, from a batch of training data, via,

$$\boldsymbol{\theta}_k \leftarrow \boldsymbol{\theta}_{k-1} - \alpha_k g_k(\boldsymbol{\theta}_{k-1}|\mathcal{E}_k), \tag{18}$$

where $\mathcal{E}_k$ denotes the random variable that decides the noise of the stochastic gradient.

It is unsurprising that no algorithm can achieve a reasonable performance if $l_k(\cdot)$ can change arbitrarily between consecutive iterations, since we can change the training data in a completely adversarial fasion. One reasonable assumption is that $l_k(\cdot)$ does not change significantly between consecutive time steps. For example, when we want to optimize information retention, the training data pool $\mathbf{S}_t$ does not change significantly between consecutive iterations. Given this intuition, we formalize our assumptions in the non-stationary case, with the definition of the mean operation $\mathbb{E}$ (which is omitted in the main paper) as follows:

**A1** Lipschitz Smoothness with constant $L \geq 0$: $\|\nabla l_k(\boldsymbol{\theta}) - \nabla l_k(\boldsymbol{\theta}')\| \leq L \|\boldsymbol{\theta} - \boldsymbol{\theta}'\|, \forall \boldsymbol{\theta}, \boldsymbol{\theta}' \in \mathbb{R}^d$ and $k$.

**A2** Unbiased gradient estimation: $\mathbb{E}_{\mathcal{E}}[g_k(\boldsymbol{\theta}|\mathcal{E})] = \nabla l_k(\boldsymbol{\theta}), \forall \boldsymbol{\theta} \in \mathbb{R}^d$ and $k$.

**A3** Bounded gradient noise with constant $\rho \geq 0$: $\|\nabla l_k(\boldsymbol{\theta}) - g_k(\boldsymbol{\theta}|\mathcal{E})\|^2 \leq \rho^2, \forall \boldsymbol{\theta} \in \mathbb{R}^d$ and $k$.

**A4** Bounded non-stationarity with constants $\chi_k \geq 0$: $|l_k(\boldsymbol{\theta}) - l_{k+1}(\boldsymbol{\theta})| \leq \chi_k, \forall \boldsymbol{\theta} \in \mathbb{R}^d$ and $k$.

Intuitively, **A1** to **A3** extend **A5** to **A7** to non-stationary objectives, and **A4** constraints the change of the objective function value between consecutive iterations, which is similar to the analysis of SGD in the convex case (Besbes et al., 2015). Note that $l_k(\boldsymbol{\theta})$ is computed over the replay data, i.e., not on the discarded historical data when the replay buffer/storage is limited. Hence the analysis in this section naturally includes limited storage as a special case. Under these assumptions, we can prove the following convergence result of SGD in the non-stationary case:

**Theorem 1.** *Assume **A1** to **A4**, and let $\alpha_k < \frac{L}{2}$,*

$$\min_{j \in \{0,1,\dots,k\}} \mathbb{E}_{\mathcal{E}_1,\dots,\mathcal{E}_j}[\|\nabla l_{j+1}(\boldsymbol{\theta}_j)\|^2] \leq T_1 + T_2 + T_3 \tag{19}$$

*where* $T_1 = \frac{2(l_1(\boldsymbol{\theta}_0) - \mathbb{E}_{\mathcal{E}_1,\dots,\mathcal{E}_{k+1}}[l_{k+2}(\boldsymbol{\theta}_{k+1})])}{\sum_{j=0}^{k}(2\alpha_{j+1} - L\alpha_{j+1}^2)}$, $T_2 = \frac{L\rho^2 \sum_{j=0}^{k} \alpha_{j+1}^2}{\sum_{j=0}^{k}(2\alpha_{j+1} - L\alpha_{j+1}^2)}$, *and* $T_3 = \frac{2\sum_{j=0}^{k} \chi_{j+1}^2}{\sum_{j=0}^{k}(2\alpha_{j+1} - L\alpha_{j+1}^2)}$.

*Proof.* With **A1** and Lemma 1.2.3 of (Nesterov, 1998), we have

$$\mathbb{E}_{\mathcal{E}_1,\ldots,\mathcal{E}_{k+1}}[l_{k+1}(\boldsymbol{\theta}_{k+1})] \tag{20}$$

$$\leq \mathbb{E}_{\mathcal{E}_1,\ldots,\mathcal{E}_{k+1}}[l_{k+1}(\boldsymbol{\theta}_k) + \nabla l_{k+1}(\boldsymbol{\theta}_k)^T(\boldsymbol{\theta}_{k+1} - \boldsymbol{\theta}_k) + \frac{L\|\boldsymbol{\theta}_{k+1} - \boldsymbol{\theta}_k\|^2}{2}] \tag{21}$$

$$= \mathbb{E}_{\mathcal{E}_1,\ldots,\mathcal{E}_k}[l_{k+1}(\boldsymbol{\theta}_k)] - \alpha_{k+1}\mathbb{E}_{\mathcal{E}_1,\ldots,\mathcal{E}_k}[\|\nabla l_{k+1}(\boldsymbol{\theta}_k)\|^2] + \frac{L\mathbb{E}_{\mathcal{E}_1,\ldots,\mathcal{E}_{k+1}}[\|\alpha_{k+1}g_{k+1}(\boldsymbol{\theta}_k|\mathcal{E}_{k+1})\|^2]}{2} \tag{22}$$

$$= \mathbb{E}_{\mathcal{E}_1,\ldots,\mathcal{E}_k}[l_{k+1}(\boldsymbol{\theta}_k)] - \alpha_{k+1}\mathbb{E}_{\mathcal{E}_1,\ldots,\mathcal{E}_k}[\|\nabla l_{k+1}(\boldsymbol{\theta}_k)\|^2] +$$
$$\frac{L\alpha_{k+1}^2 \mathbb{E}_{\mathcal{E}_1,\ldots,\mathcal{E}_{k+1}}[\|(g_{k+1}(\boldsymbol{\theta}_k|\mathcal{E}_{k+1}) - \nabla l_{k+1}(\boldsymbol{\theta}_k)) + \nabla l_{k+1}(\boldsymbol{\theta}_k)\|^2]}{2} \tag{23}$$

$$= \mathbb{E}_{\mathcal{E}_1,\ldots,\mathcal{E}_k}[l_{k+1}(\boldsymbol{\theta}_k)] - \alpha_{k+1}\mathbb{E}_{\mathcal{E}_1,\ldots,\mathcal{E}_k}[\|\nabla l_{k+1}(\boldsymbol{\theta}_k)\|^2] +$$
$$L\alpha_{k+1}^2 \mathbb{E}_{\mathcal{E}_1,\ldots,\mathcal{E}_{k+1}}[\|g_{k+1}(\boldsymbol{\theta}_k|\mathcal{E}_{k+1}) - \nabla l_{k+1}(\boldsymbol{\theta}_k)\|^2 + \|\nabla l_{k+1}(\boldsymbol{\theta}_k)\|^2 -$$
$$\nabla l_{k+1}(\boldsymbol{\theta}_k)^T(g_{k+1}(\boldsymbol{\theta}_k|\mathcal{E}_{k+1}) - \nabla l_{k+1}(\boldsymbol{\theta}_k))]/2 \tag{24}$$

$$\leq \mathbb{E}_{\mathcal{E}_1,\ldots,\mathcal{E}_k}[l_{k+1}(\boldsymbol{\theta}_k)] - \alpha_{k+1}\mathbb{E}_{\mathcal{E}_1,\ldots,\mathcal{E}_k}[\|\nabla l_{k+1}(\boldsymbol{\theta}_k)\|^2] + \frac{L\alpha_{k+1}^2 \mathbb{E}_{\mathcal{E}_1,\ldots,\mathcal{E}_k}[\rho^2 + \|\nabla l_{k+1}(\boldsymbol{\theta}_k)\|^2]}{2} \tag{25}$$

$$= \mathbb{E}_{\mathcal{E}_1,\ldots,\mathcal{E}_k}[l_{k+1}(\boldsymbol{\theta}_k)] - (\alpha_{k+1} - \frac{L\alpha_{k+1}^2}{2})\mathbb{E}_{\mathcal{E}_1,\ldots,\mathcal{E}_k}[\|\nabla l_{k+1}(\boldsymbol{\theta}_k)\|^2] + \frac{L\alpha_{k+1}^2 \rho^2}{2}. \tag{26}$$

where (25) is due to **A3** ($\|g_{k+1}(\boldsymbol{\theta}_k|\mathcal{E}_{k+1}) - \nabla l_{k+1}(\boldsymbol{\theta}_k)\|^2 \leq \rho^2$) and **A2** ($\mathbb{E}_{\mathcal{E}_1,\ldots,\mathcal{E}_{k+1}}[-2\nabla l_{k+1}(\boldsymbol{\theta}_k)^T(g_{k+1}(\boldsymbol{\theta}_k|\mathcal{E}_{k+1}) - \nabla l_{k+1}(\boldsymbol{\theta}_k))] = 0$). Rearranging the terms in (26) and using **A4**, we have

$$(\alpha_{k+1} - \frac{L\alpha_{k+1}^2}{2})\mathbb{E}_{\mathcal{E}_1,\ldots,\mathcal{E}_k}[\|\nabla l_{k+1}(\boldsymbol{\theta}_k)\|^2] \tag{27}$$

$$\leq \mathbb{E}_{\mathcal{E}_1,\ldots,\mathcal{E}_k}[l_{k+1}(\boldsymbol{\theta}_k)] - \mathbb{E}_{\mathcal{E}_1,\ldots,\mathcal{E}_{k+1}}[l_{k+1}(\boldsymbol{\theta}_{k+1})] + \frac{L\alpha_{k+1}^2 \rho^2}{2}, \tag{28}$$

$$\leq \mathbb{E}_{\mathcal{E}_1,\ldots,\mathcal{E}_k}[l_{k+1}(\boldsymbol{\theta}_k)] - \mathbb{E}_{\mathcal{E}_1,\ldots,\mathcal{E}_{k+1}}[l_{k+2}(\boldsymbol{\theta}_{k+1})] + \chi_{k+1} + \frac{L\alpha_{k+1}^2 \rho^2}{2}. \tag{29}$$

Summing both sides of (29) from $j = 0$ to $k$, we have

$$\sum_{j=0}^{k}(2\alpha_{j+1} - L\alpha_{j+1}^2)\mathbb{E}_{\mathcal{E}_1,\ldots,\mathcal{E}_j}[\|\nabla l_{j+1}(\boldsymbol{\theta}_j)\|^2]$$

$$\leq 2(l_1(\boldsymbol{\theta}_0) - \mathbb{E}_{\mathcal{E}_1,\ldots,\mathcal{E}_{k+1}}[l_{k+2}(\boldsymbol{\theta}_{k+1})]) + L\rho^2 \sum_{j=0}^{k}\alpha_{j+1}^2 + \sum_{j=0}^{k}\chi_{j+1}. \tag{30}$$

With a similar derivation as in the proof of Theorem 2, we immediately have (19). □

Intuitively, to make $T_3$ converge to 0, we need to set the learning rate such that

$$\lim_{k \to \infty} \frac{\sum_{j=0}^{k} \chi_{j+1}}{\sum_{j=0}^{k}(2\alpha_{j+1} - L\alpha_{j+1}^2)} = 0, \tag{31}$$

i.e., we need $\sum_{j=0}^{k}(2\alpha_{j+1} - L\alpha_{j+1}^2)$ to increase faster than $\sum_{j=0}^{k}\chi_{j+1}$. In practice, the value of $\chi_k$ depends on the application. Given a constant learning rate $\alpha_k = \alpha$, $T_3$ can increase for an increasing sequence of $\chi_k$. This explains why **P3** can happen in practice. If $\chi_k \geq \chi$ for a constant $\chi$, we cannot have a learning rate schedule that makes $T_1$, $T_2$ and $T_3$ converge to 0 at the same time. Moreover, when $\alpha_k < 1$, decreasing $\alpha_k$ will increase of $T_3$. In the extreme case, if $\alpha_k$ is reduced to 0, $T_3$ will increase in linear speed, causing long term failure of SGD+RWP. This explains **P2**, i.e., why reducing the learning rate to 0 using RWP hurts the performance of OCL. And demonstrates why we need to keep the learning rate sufficiently large, i.e., **P1**. Note that currently, there is no lower bound to demonstrate the tightness of (19). Hence, the analysis in this section can only be viewed as an intuition.

**Interpreting Theorem 1 under limited storage** In practice, both supervised learning and continual learning use a limited amount of (though may be in a large scale) training/replay data. In such cases,

the loss ($l_k(\boldsymbol{\theta})$ in (17) and $l(\boldsymbol{\theta})$ (10)) is only computed over the stored data. And our assumptions in the theories are only w.r.t. the loss over the training/replay data. For supervised learning, each training data point can be viewed as a random sample from the original data distribution. For continual learning where not all past data can be stored, one can construct a reservoir replay buffer (Chaudhry et al., 2019) online, so that at each time step $t$, each data point in the replay buffer is also a random sample from the original data distribution. Hence, the bound in Theorem 1 is directly comparable to the bound in Theorem 2 when the replay buffer size in OCL is the same as the dataset size of supervised learning. The performance of a (continual) learner over the original data distribution is the problem of generalization, which is out of the scope of convergence analysis.

## B   FURTHER DETAILS OF AMA AND ONLINE VALIDATION

---

**Algorithm 2** Adaptive Moving Average (AMA)

---

**Require:** Initial model $\boldsymbol{\theta}_0$, the initial moving average weight $\gamma_0$, MA weight adjustment coefficient $\delta(>0)$, model update interval $K_M$, validation interval $K_V$, MA weight update interval $K_W$, number of training iterations per time step $p$.

1: $\boldsymbol{\theta}^{MA_1} \leftarrow \boldsymbol{\theta}_0, \boldsymbol{\theta}^{MA_2} \leftarrow \boldsymbol{\theta}_0, \gamma^{MA_1} \leftarrow \gamma_0, \gamma^{MA_2} \leftarrow \frac{\gamma_0}{\delta}$

2: $Acc_1 \leftarrow 0, Acc_2 \leftarrow 0, n \leftarrow 0, i_{best} \leftarrow 1$

3: **for** $k \in \{1, 2, 3, ..., \infty\}$ **do**

4:     $\boldsymbol{\theta}_k \leftarrow \boldsymbol{\theta}_{k-1} - \alpha_k g(\boldsymbol{\theta}_{k-1})$

5:     **if** $k \mod K_M = 0$ **then**

6:         $\boldsymbol{\theta}^{MA_1} \leftarrow \gamma^{MA_1}\boldsymbol{\theta}^{MA_1} + (1 - \gamma^{MA_1})\boldsymbol{\theta}_k$

7:         $\boldsymbol{\theta}^{MA_2} \leftarrow \gamma^{MA_2}\boldsymbol{\theta}^{MA_2} + (1 - \gamma^{MA_2})\boldsymbol{\theta}_k$

8:     **end if**

9:     **if** $k \mod K_V = 0$ **then**

10:         $Acc_1 \leftarrow \frac{n \cdot Acc_1 + \text{acc}(\mathbf{B}^V_{\lfloor \frac{k}{p} \rfloor}, \boldsymbol{\theta}^{MA_1})}{n+1}, Acc_2 \leftarrow \frac{n \cdot Acc_2 + \text{acc}(\mathbf{B}^V_{\lfloor \frac{k}{p} \rfloor}, \boldsymbol{\theta}^{MA_2})}{n+1}, n \leftarrow n+1$

11:         **If** $Acc_1 > Acc_2$ **then** $i_{best} = 1$ **else** $i_{best} = 2$

12:     **end if**

13:     **if** $k \mod K_W = 0$ **then**

14:         $Acc_1 \leftarrow 0, Acc_2 \leftarrow 0, n \leftarrow 0$

15:         **if** $i_{best} = 1$ **then**        ▷ Increase both MA weights but limit them to be $\leq 1$

16:             $\gamma^{MA_1} \leftarrow \min(1, \delta\gamma^{MA_1}), \gamma^{MA_2} \leftarrow \frac{\gamma^{MA_1}}{\delta}$

17:             $\boldsymbol{\theta}^{MA_2} \leftarrow \boldsymbol{\theta}^{MA_1}, i_{best} = 2$

18:         **else**                 ▷ Decrease both MA weights

19:             $\gamma^{MA_1} \leftarrow \frac{\gamma^{MA_1}}{\delta}, \gamma^{MA_2} \leftarrow \frac{\gamma^{MA_2}}{\delta}$

20:             $\boldsymbol{\theta}^{MA_1} \leftarrow \boldsymbol{\theta}^{MA_2}, i_{best} = 1$

21:         **end if**

22:     **end if**

23: **end for**

---

Alg. 2 describes the complete version of Alg. 1, with overhead reduction strategies added. Each online validation step (Line 9 to Line 12) computes the accuracy on a batch of data $\mathbf{B}^V_{\lfloor \frac{k}{p} \rfloor}$ sampled from the information retention validation set $\mathbf{S}^V_{\lfloor \frac{k}{p} \rfloor}$ of the current training iteration $k$, where $p$ is the number of SGD updates allowed per time step. We average the computed accuracy on the fly between each $K_W$ iterations, and use the averaged accuracy as the online information retention validation accuracy. The benefit of this online validation process is that the validation steps can be parallelized completely with model updates, and we can also select the best MA model for inference anytime during OCL.

## C   FURTHER EXPERIMENTS

### C.1   BN VS GN+WS

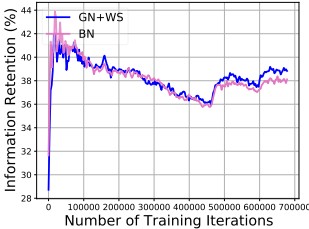

Figure 6: **Information retention over time for models with different normalization layers.** (↑) GN+WS performed slightly better than BN in terms of information retention, but more efficient in AMA due to the removal of BN statistics.

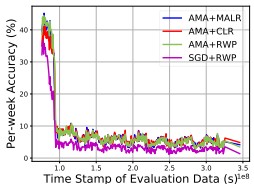
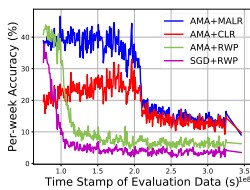

(a) $k = 5.76e5$. $\alpha_k$ is $1.25e-2$ for AMA+MALR, $6.25e-3$ for AMA+RWP, and $9.77e-5$ for SGD+RWP. (↑)

(b) $k = 4.8e8$. $\alpha_k$ is $1.56e-3$ for AMA+MALR, $9.54e-8$ for AMA+RWP, and $2.98e-9$ for SGD+RWP. (↑)

Figure 7: **Accuracy on evaluation data from different time ranges, for models at two different training iterations** $k$. Comparing to AMA+MALR, AMA+CLR used a larger learning rate $\alpha_k$, causing performance deterioration on all data ranges. SGD+RWP and AMA+RWP dropped the learning rate too quickly, making the model unable to adapt to new data.

As mentioned in Sec. 4.2, we replace BN with GN+WS to remove the overhead of re-computing BN statistics in AMA. To show the effect of changing the normalization layer, we compare the model trained with BN against the one trained with GN+WS. Due to resource limitations, we only train the BN model for 700 thousand iterations. As shown in Fig. 6, GN+WS performed slightly better than BN. Combined with the efficiency, GN+WS is suitable for information retention in OCL.

## C.2 Small Learning Rates Stymie Adaptation to New Data

To understand the effect of using suboptimal learning rates on data from different ranges, we take the models of different methods at two distinct training iterations ($k = 5.76e5$ and $k = 4.8e8$), and plot their accuracy on evaluation data from different time ranges. As shown in Fig. 7, AMA+CLR performed worse than AMA+MALR on all data ranges due to the use of a large learning rate. The performance gap was larger on historical data than on future data. SGD+RWP and AMA+RWP performed well only on early data and failed to improve the performance for new data when the learning rate was too low.

## C.3 Heavy parallelism Hurt OCL

Cai et al. (2021) have shown that increasing batch sizes for optimizing learning efficacy hurts all OCL performance metrics. We perform a similar analysis to study whether this phenomenon still happens when optimizing information retention. Specifically, we optimize information retention with batch sizes varying from 64 to 1024. When we increase the batch size, we reduce the number of training iterations per time step and increase the learning rate by the same factor. We allow 1 training iteration per time step for a batch size of 256. As shown in Fig. 8, increasing batch sizes for optimizing information retention also hurt all OCL performance metrics.

## C.4 A Reasonably Large Replay Buffer Is Sufficient

We now analyze the effect of replay buffer sizes, by training models with replay buffer sizes of 400 thousand, 4 million, and 40 million images. To optimize information retention, we use a reservoir

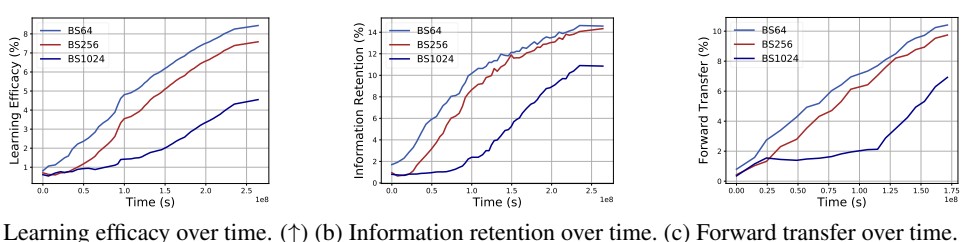

(a) Learning efficacy over time. (↑) (b) Information retention over time. (c) Forward transfer over time. (↑)
(↑)

Figure 8: **Effect of batch sizes.** Increasing the batch size hurt all OCL performance metrics.

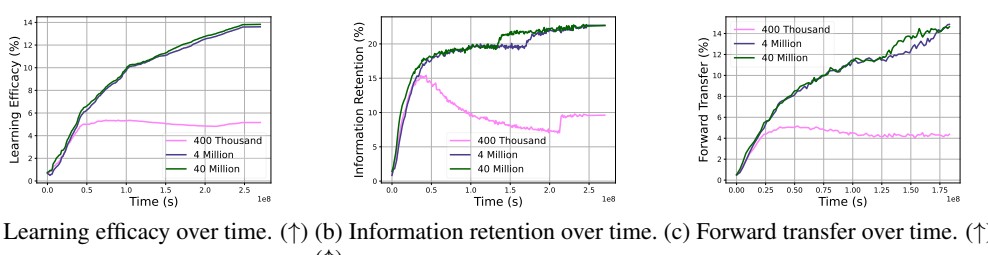

(a) Learning efficacy over time. (↑) (b) Information retention over time. (c) Forward transfer over time. (↑)
(↑)

Figure 9: **Effect of replay buffer sizes.** Though all performance metrics dropped when the replay buffer size was reduced to 400 thousand images, the performance remained similar when the replay buffer size was larger than 4 million images.

buffer (Chaudhry et al., 2019) in this experiment. We allow 5 training iterations per time step for each replay buffer size.

As shown in Fig. 9, reducing the replay buffer size from 40 million to 400 thousand images significantly hurt all performance metrics. However, the performance remained similar when the replay buffer size was 4 million or larger. Therefore, given a fixed model and computational budget, OCL only requires a *reasonably large* replay buffer.

### C.5   APPLY AMA TO OTHER BASE OPTIMIZERS

Besides SGD, AMA+MALR can also be combined with other base optimizers. As an example, we apply it to ADAM and test the performance on CLOC. Specifically, we compare (ADAM-based) AMA+MALR with ADAM+MALR and ADAM+RWP, with an initial learning rate tuned to 0.001. Due to resource limitations, we only run all algorithms on partial data. As shown in Fig. 10, (ADAM) AMA+MALR outperformed ADAM+RWP. However, ADAM methods performed worse than SGD. We conjecture that the adaptive scaling in ADAM reduced the effective learning rate (similar to the SGD+RWP case), and leave further investigations as future works. Due to the poor performance of ADAM-based methods, we use SGD-based methods in the main experiments.

### C.6   IMPACT OF OPTIMIZATION OBJECTIVES TO OCL

An ideal OCL algorithm should perform well for all three metrics, i.e., learning efficacy (eq. (1)), information retention (eq. (2)), and forward transfer (eq. (3)). However, optimizing one objective can hurt others. To demonstrate this point, we compare the model trained to optimize information retention (with AMA+MALR), and the model trained to optimize learning efficacy (with the method of Cai et al. (2021)). Following Cai et al. (2021), we allow different computation budgets (1 and 5 training iterations) per time step. To fairly compare pure replay and mixed replay, we double the batch size for AMA+MALR, so that the average number of gradient descent steps per image is the same for the two models. We set $k_1$ and $k_2$ in the forward transfer metric to $10\%$ and $25\%$ of the dataset size respectively.

As shown in Fig. 11a, given the same budget, optimizing learning efficacy (LE Model) performed the best for learning efficacy. Optimizing information retention (IR Model) performed the best for information retention (Fig. 11b) and forward transfer (Fig. 11c). More importantly, unlike optimizing

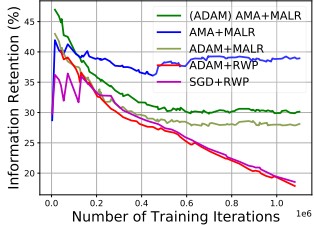

Figure 10: **ADAM results**. (↑) AMA+MALR applied to ADAM also improved information retention in OCL. However, ADAM-based methods performed worse than SGD counterparts.

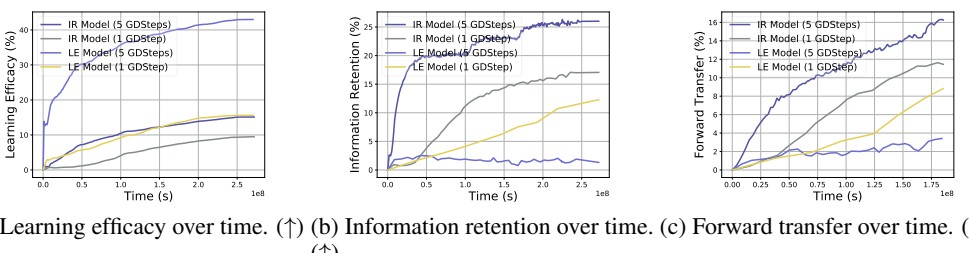

(a) Learning efficacy over time. (↑) (b) Information retention over time. (c) Forward transfer over time. (↑) (↑)

Figure 11: **Optimizing learning efficacy (LE Model) vs optimizing information retention (IR Model).** Optimizing information retention hurt learning efficacy (a), but improved information retention (b) and forward transfer (c). More importantly, increasing the budget for optimizing learning efficacy hurt both information retention and forward transfer. In contrast, increasing the budget for optimizing information retention improved all OCL metrics.

information retention, where increasing the computation budget improved all OCL metrics, increasing the computation budget for optimizing learning efficacy *hurt* information retention and forward transfer. This phenomenon is consistent with the finding in the language modeling setting (Hu et al., 2020). Hence in terms of improving long term transfer in OCL, information retention is a better objective than learning efficacy.

## C.7 DETAILS ON GOOGLE LANDMARKS V2

Similar to CLOC, we construct the OCL version of Google Landmarks v2 using the subset of images from Flickrs, which have time stamps that can be retrieved from flickr API. Similar to the original paper (Weyand et al., 2020), we address the noisy label and class imbalance issues by using only the data from the cleaned set, and filter out the classes with fewer than 25 images. This results in 645666 images in total. We further divide the dataset into $90\%$ of training data (roughly 580 thousand images) and $10\%$ of evaluation data (roughly 65 thousand images).

In terms of hyper-parameters, we use the same model as the CLOC dataset. We tune other hyper-parameters on the first $20\%$ of the training data (we still use this part of data during training because otherwise the dataset will be even smaller), resulting in the learning rate of 0.135, the momentum of 0.9 for SGD, the weight decay of 1e-4, and the cross-entropy loss.

## C.8 DETAILS ON IMAGENET CONTINUAL LEARNING EXPERIMENTS

We mostly follow the hyper-parameter settings of supervised learning, i.e., we use ResNet50, with batch size of 256, initial learning rate of 0.1, the momentum of 0.9 for SGD, the weight decay of 1e-4. For ER, GDumb and MIR, we use the same basic hyper-parameter as SGD+Cyclic, including the optimizer, learning rates, the batch size etc. For ER, we mix 128 samples from the current task and 128 samples from replay for each model udpate step. For MIR, we select 128 examples from 256 candidates (using the $s_{MI-1}$ criterion) during the replay data sampling procedure, and mix them with another 128 examples from the current task to update the model. For a fair comparison, we limit the computation of each method to be the same as SGD+Cyclic. Specifically, MIR requires at least two SGD iterations in each model update step, hence, the total number of model update steps is limited to

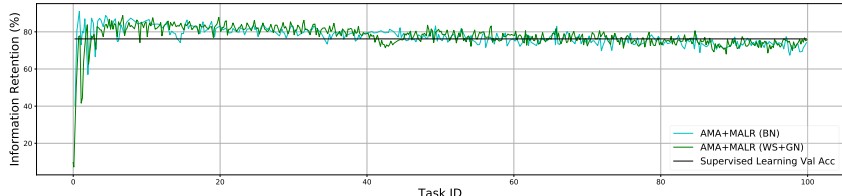

Figure 12: **BN vs GN+WS in ImageNet continual learning.** (↑) Similar as in CLOC, GN+WS also performed slightly better than the BN model on ImageNet continual learning. And its best validation accuracy in the final task (76.8%) is similar as the standard BN model trained with supervised learning.

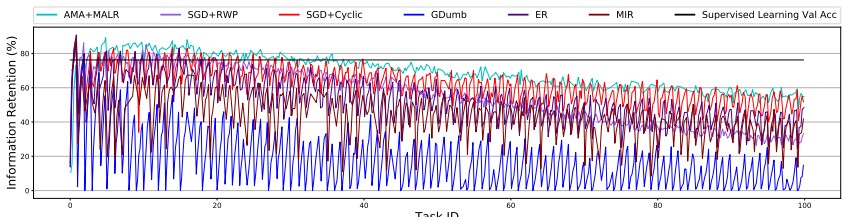

Figure 13: **Information retention on ImageNet continual learning with small replay buffer (120K images).** (↑) Similar to the results in the main experiment, AMA+MALR out-performed SGD+Cyclic and other OCL methods designed for handling storage limitations. Interestingly, the gap between SGD+Cyclic and ER/MIR became smaller in this case.

half of the SGD steps in SGD+Cyclic. GDumb trains a model from scratch for each task, hence the number of SGD updates of GDumb and SGD+Cyclic is limit to be the same in each task. To make the final validation accuracy directly comparable to the standard ResNet50 supervised learning result, we use BN as the normalization layer in this experiment (The GN+WS model performed slightly better than the BN model). To update BN statistics on the fly, we do one extra forward pass per 10 training iterations to accumulate the BN statistics for MA models. For both RWP and AMA, we decay the learning rate by half when the corresponding conditions are satisfied. We set $K_R$ in RWP and AMA to 15000, which is smaller than in CLOC, so that the effect of learning rate reduction can be clearly seen.

We also show in Fig. 12 the performance comparison between the BN model and the GN+WS model. Similar as in CLOC, GN+WS also performed slightly better than BN. And its best validation accuracy in the final task (76.8%) is similar as the standard BN model trained with supervised learning. This result confirmed the generalization of our method across problems and datasets.

In the main paper, we analysed the complexity of AMA. To support our analysis, we further provide here the runtime of each algorithm to show the practical efficiency of AMA. Specifically, we provide the average runtime for each algorithm to finish one training iteration. AMA+MALR used 0.264s, SGD+Cyclic used 0.245s, SGD+RWP used 0.272s, GDumb used 0.261s, MIR used 0.395s, ER used 0.327s. Note that there are some noise in the runtime due to the change of environments in different experiment executions, but it is clear that AMA+MALR has similar efficiency as SGD+RWP in practice.

### C.9 SMALL REPLAY BUFFER ON IMAGENET

In this section, we show the ImageNet continual learning result with limited storage. Specifically, we only allow 120K images to be stored in the replay buffer, and compare AMA+MALR against SGD+Cyclic, ER, MIR and GDumb. For AMA+MALR, SGD+Cyclic and GDumb, the replay buffer is formed using reservoir sampling. For ER and MIR, we assign half of the replay buffer for storing data from previous tasks (using reservoir sampling), and half of the replay buffer for storing data from the current task. As shown in Fig. 13, AMA+MALR still outperformed all competitors in this case. Interestingly, the performance gap between SGD+Cyclic and ER/MIR became smaller compared to the case of large replay buffers.

## C.10 Performance on Supervised Learning

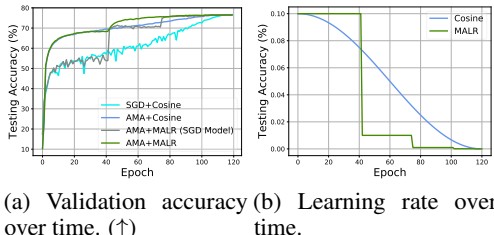

(a) Validation accuracy over time. (↑)  (b) Learning rate over time.

Figure 14: **Performance of AMA and MALR on ImageNet for supervised learning.** (↑) Similar as in the OCL case, AMA performed much better than SGD when the learning rate was large. MALR adaptively reduced the learning rate, providing similar final performance as the cosine schedule.

To understand the performance of AMA + MALR on supervised learning, we apply it to ImageNet, and compare against SGD + cosine schedule (denoted as SGD+Cosine), which is the standard for ImageNet. To understand the effect of each component, we also show the performance of AMA+Cosine and the SGD model in AMA+MALR. To replicate the standard ResNet50 result on ImageNet, we use BN as the normalization layer, and train all models for 120 epochs. Since limiting the minimum learning rate hurts supervised learning, we only use **C1** and **C2** in MALR for this experiment. We reduce the learning rate by 10 times once **C1** and **C2** are true.

As shown in Fig. 14, though not better in terms of the final performance, AMA performed much better than SGD at early training stages, where the learning rate was large. MALR adaptively reduced the learning rate over time, providing a similar final performance as the cosine schedule.

## C.11 AMA hyper-parameter tuning and sensitivity

As described in Sec. 4.2, we set the MA weight adjustment coefficient $\delta$ to 5 and the interval $K_W$ to adjust MA weights to 10000 iterations. We empirically observed that setting $K_W$ in between 1-10000, and $\delta$ between 2-10 does not have strong effect to the performance (i.e., the classification accuracy). Setting $K_W$ to 10000 and $\delta$ to 5 worked across problems (ImageNet, CLOC, Google Landmarks) and provides AMA a similar speed as SGD in practice.

For classification tasks, we set $\epsilon$ in **C3** to $3\%$ accuracy. This value worked for CLOC, ImageNet and CGLM (only tuned on the CLOC preprocessing data). It can be used as a reasonable default for other datasets as well. For tasks other than classification, one can tune $\epsilon$ by:

1. Finding the best initial learning rate on preprocessing data.

2. Using a constant learning rate schedule, with learning rate set to the best initial learning rate. Compute the performance gap between AMA and SGD models on pre-processing data, denoted as g.

3. Setting $\epsilon$ to roughly 0.3g (this is how we set our $\epsilon$), we observe that the performance difference is small by setting $\epsilon$ in between 0.5-0.2g.

## C.12 Why not smaller datasets like CIFAR or 1 training iteration per time step?

Existing literature focus more on datasets with smaller scale, e.g., CIFAR, MNIST and/or with 1 training iteration per time step. Here we show why such experiments are not considered in this work. As shown in Fig. 15 if we allow only 1 iteration per time step for CIFAR (5 tasks, 2 classes for each task) and ImageNet, the learning rate for both SGD+RWP and AMA+MALR would remain constant throughout training, hence, no learning rate scheduling is needed. Also as shown in Fig. 16, even if we allow 120 iterations per time step in CIFAR, as in the ImageNet experiment, SGD+RWP and AMA+MALR still used constant learning rate throughout training, as in the experiment of CGLM (Sec. 5.2.1). More importantly, the performance difference between different continual learning algorithms was much smaller in CIFAR compared to ImageNet, due to the small data stream. Hence, we argue that the data stream in such cases is too short for the main purpose of this work, which is to

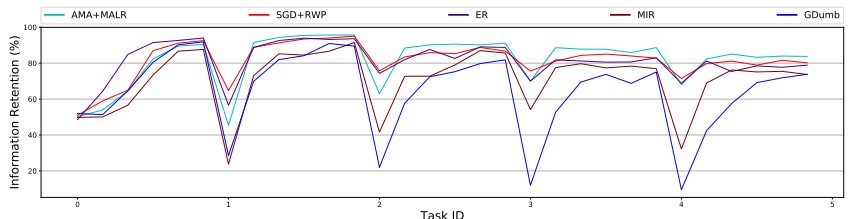

Figure 15: **Information retention on ImageNet (Left) and CIFAR (right) continual learning with 1 training iteration per time step.** (↑) We store 120K images in the replay for ImageNet and 5K for CIFAR. Due to the small number of training iterations in total, allowing 1 training iteration per time step did not reveal the problem of SGD+RWP in the long term. Specifically, the learning rate of SGD+RWP and AMA+MALR remained constant throughout training for both CIFAR and ImageNet. And the advantage of AMA+MALR was also invisible since MA needs a reasonable number of iterations to track the SGD trajectory. Hence, this experiment is not proper to validate Theorem 7 and demonstrate the complete behavior of different algorithms in large scale OCL.

Figure 16: **Information retention on CIFAR continual learning with 120 training iteration per time step, with replay buffer size of 5000 images.** (↑) Similar as in the experiment of CGLM (Sec. 5.2.1), due to the small number of images in total, training on CIFAR does not reveal the problem of SGD+RWP in the long term even with *multiple* training iterations per time step. The learning rate of SGD+RWP and AMA+MALR remained constant throughout training, and the performance of different algorithms was much smaller than in the ImageNet experiment. Hence, this experiment is also not proper to validate Theorem 7 and demonstrate the complete behavior of different algorithms in the long term.

reveal the problem of SGD+RWP (validating Theorem 7) and the benefit of AMA+MALR in the long term.

