# OpenReview forum: "Improving Information Retention in Large Scale Online Continual Learning"
_ICLR.cc/2023/Conference — Submitted to ICLR 2023_

### Official Review · Reviewer_u5yA · 2022-10-23

**Confidence:** 4
**Correctness:** 2
**Technical Novelty And Significance:** 2
**Empirical Novelty And Significance:** 2
**Recommendation:** 3

**Clarity, Quality, Novelty And Reproducibility:**

Mostly clear
No code uploaded.

**Strength And Weaknesses:**

Strength
Studying different training methods is interesting in online continual learning.

Weaknesses
My understanding of information retention is to avoid catastrophic forgetting. Why not use the standard terminology?

Online continual learning is a non-stop process, I don’t understand what you mean by convergence. In standard online continual learning, whenever a small batch of data from the stream is collected it is trained in one iteration. New tasks of completely different distributions (with new class labels) come constantly. In such a non-stationary data setting, it is unclear how you know when the convergency should happen.

You wrote “We allow 80 training iterations per time step,” but that is not the standard online continual learning setting. My understanding is that you are doing online learning rather than online continual learning. In online continual learning, it is assumed that the system sees the data only once. Once a small batch of training data comes it is immediately trained in one iteration. Some classes as a task may come early and some classes may come later in a sequence of tasks. So, the distribution change at the task switch is drastic and the time of each such switch is unknown. I don't think that you are working under this setting, but I am not familiar with the datasets that you are using. In the standard online continual learning setting, it is unclear how your adaptive training can be applied.

Again, I think you are working on online/incremental learning rather than online continual learning. Then, you should compare with methods in online learning literature. I'm not working in that field, so I cannot give you any references off the top of my head. But I did read some papers in the area a while ago.

Since your data is not commonly used in online continual learning, it is hard for me to see how your method performs against other methods. There are many online continual learning benchmark datasets. I think you should compare with those existing online continual learning baselines using the existing benchmarks. I am surprised that you did not compare with any existing online continual learning baselines, but only various variants of training methods. If no comparison, I won’t know that your information retention is better. For example, some recent systems are (you can find a lot of other baselines in these papers):
1. Mai et al. Supervised contrastive replay: Revisiting the nearest class mean classifier in online class-incremental continual learning. CVPR workshop’2021.
2. Bang et al. Rainbow memory: Continual learning with a memory of diverse samples. CVPR’2021.
3. Guo et al. Online Continual Learning through Mutual Information Maximization, ICML’2022.

If you are working under a different setting, you should have compared with the existing setting and stated the differences. The existing online continual learning systems should be easily adapted for your setting by simply learning one task of all classes. Actually, in the case, existing online learning methods may do better.

It is unclear whether you are using your adaptive training schedule at each time step or in the time span of the whole training process. My guess it is the latter.

The figures in Figure 1 are too small to see.


**Summary Of The Paper:**

This paper studies various adaptive training methods and schedules in online continual learning.

**Summary Of The Review:**

I think the paper is studying online learning rather than online continual learning. It didn't compare with any existing online learning or online continual learning methods. The adaptive learning methods and schedules proposed in the paper are not  new as they have been used in other areas.

---

> ### Author Response · Authors · 2022-11-11
> **Thanks for the valuable comments. We believe most of your concerns are due to misunderstandings, which address below. We kindly ask you to reconsider your opinion and scoring.**
>
>  ## 1) Why use information retention rather than catastrophic forgetting.
>
> Information retention refers to *the mathematically defined (Eq. (2)) metric of maintaining previous knowledge*. Catastrophic forgetting is the problem, and ``solving catastrophic forgetting” is not compact enough and is not a good name for a formal metric. In the meantime, catastrophic forgetting is mostly used in the case of limited memory, which provides the impression that the performance drop on historical samples are due to insufficient memory. We show here that information retention is still a problem even when the memory is sufficient to store all past data. The goal of this paper is to demonstrate that at least part of the problem for information retention lies in optimization, not the memory limitation. Finally, information retention is also used in recent large scale OCL benchmark [1], which we consistently use since we evaluated our method on the same benchmark.
>
> [1] Online continual learning with natural distribution shifts: An empirical study with visual data.
>
>
>
> ## 2) Notion of convergence in OCL.
> Though OCL is a non-stop process, there is a notion of convergence since our optimization process is also non-stopping. Our notion of convergence follows online optimization [2], the notion of convergence means that if online continual learning is successful, i.e., an algorithm is effective, the average error over time should decrease and remain low over the non-stop process. And the average error over time has a proxy of the average gradient norm, which is used to analyze the convergence of SGD in OCL in theorem 1.
>
> Another reason to use convergence is because the bound of (7) is directly comparable to the convergence bound of the stationary case, i.e., theorem 2 in Appendix A.1. But we never say that using the word ``convergence” means (7) will reduce to 0 by SGD, we actually show that (7) remains non-zero if distribution shifts remains over this non-stop process, which lead to the problem of SGD.
>
> [2] Elad Hazan. Introduction to online convex optimization.
>
> ## 3) You wrote “We allow 80 training iterations per time step,” but that is not the standard online continual learning setting. My understanding is that you are doing online learning rather than online continual learning.
>
> We kindly note that we are **not** doing online learning. We follow the standard OCL setting in the original CLOC paper [2], which also **allows multiple training iterations** per time step. The concept of online means you have a limited computational budget at each time step, which is far from the budget that allows you to retrain the model from scratch at each time step. The budget can allow multiple training iterations per time step, please see [1,3,4] for the same definition as ours. In all these works, the setting of 5 iterations per time step is evaluated.
>
> Also as we stated clearly in the experiments, we allow 80 rather than 1 or 5 training iterations per time step because we want to ensure a sufficiently large training iterations so that we can observe the heavy performance drop of SGD+RWP and **empirically verify theorem 1**. We did not observe a similar trend using a smaller dataset (see CGLM in Sec. 5.2.1, with only 580K training images) or just 1 training iteration per time step. To further demonstrate this point, we perform a similar ImageNet experiment as in Fig. 13, but limiting the number of training iterations per time step to be 1. We train both SGD+RWP and AMA+MALR. As shown in Fig. 15 of the revision (we upload only the figure for now, will add detailed description later), due to the small number of training steps in total, we could not observe the strong performance degration of SGD+RWP or the advantage of AMA+MALR (as in the 1-st task of Fig. 5). In fact, the learning rate of SGD+RWP and AMA+MALR remained the same as the initial value.
>
> The setting of multiple training iterations per batch is also common in practice since the speed of the data stream varies across problems. The number of iterations we can perform at each time step depends also on the model we use (e.g., given the same data stream speed, if using resnet-101 allows only 1 iteration, then using resnet-50 should allow more than 1 iterations).
>
> Also note that we defer from traditional online continual learning in the way that we **explicitly** limit the computation budget. E.g., some existing online continual learning methods (as we compared against in Sec. 5.2.1) implicitly allows more computation budgets than plain SGD by performing multiple forward and backward passes or larger batch sizes in each training iteration. For these methods, we also explicitly limit these budgets to be the same as SGD. We will state our setup clearer in the revision, and discuss its difference to existing OCL settings in the related work.
>
> [3] Online Continual Learning from Imbalanced Data.
>
> [4] Online continual learning with maximally interfered retrieval.

---

> > ### Author Response · Authors · 2022-11-11
> > **Response to other comments (separate due to space limitation).**
> >
> > ## 4) Compare with existing OCL methods.
> >
> > We kindly note that we have compared our method against existing OCL methods including the original CLOC method on CLOC [2] and GDumb [5], MIR [4], ER [6] on ImageNet.
> >
> > [5] Gdumb: A simple approach that questions our progress in continual learning.
> >
> > [6] On tiny episodic memories in continual learning.
> >
> > Please refer to Fig 5. for the case of unlimited memory and Fig. 13 for the case of limited memory. In both cases, existing OCL methods were less effective than our approach, due to the reason of explicitly limited computation (mentioned in response 3)). This supports our claim that limited computation and limited memory are two orthogonal problems, existing OCL methods designed for limited memory trade more computation for better performance, which become less effective when the memory is sufficiently large and the computation is limited.
> >
> > The reason that our main experiment compared mainly AMA+MALR with SGD+RWP is because we build AMA + MALR on top of SGD + RWP. Note that we did not apply MIR, ER or GDumb to CLOC and CGLM because these two benchmarks have smooth distribution shifts with no clear task boundaries. Standard OCL algorithms are typically applied to the case with sharp task boundaries, where they set the same learning rate schedule for each individual task, i.e., decay the learning rate from an initial value to 0 within each task. But this cannot be applied to smooth stream of CLOC and CGLM. Nonetheless, we have demonstrated that existing methods designed for addressing the memory limitation is much less effective in the case of limited computation.
> >
> >
> > ## 5) Standard continual learning benchmarks.
> > We note that we have evaluated our method on ImageNet under the standard task-incremental setup with 100 tasks, **which is the only *large scale* task-incremental dataset that we found to be able to demonstrate the poor performance of SGD+RWP and the effectiveness of AMA in the long term**. For other ``standard’’ continual learning benchmarks like CIFAR and MNIST with less than 100K images, the data stream is too short to reveal the degraded performance of different algorithms (SGD+RWP and other continual learning methods we compared against in the ImageNet experiments) in the long term. This phenomenon can also be seen from Fig. 15 mentioned in response **3)**, or Fig. 4 (where SGD+RWP did not enter the poor convergence phase given only 580K images), or from Fig. 5 and Fig. 13, where the performance between all methods were much smaller for the first  5-10 tasks (which is roughly at the size of CIFAR and MNIST). Hence we argue again that we focus on large scale continual learning where the data stream is long enough to **empirically validate theorem 1 and reveal more problems of existing algorithms**.  CIFAR and MINST datasets are not suitable for this purpose. To make this point clearer, we have added similar experiments on CIFAR in Sec. C.12, specifically in FIg. 15 (1-iter CIFAR) and Fig. 16 (120-iter CIFAR). Similar as in the experiment for CGLM, the learning rate for both SGD+RWP and AMA+MALR remained constant throughout training, and we did not observe a severe performance drop for SGD+RWP since the stream is too short. Also the performance difference between different algorithms was less obvious on CIFAR under our setup, even with 120 training iterations per time step. This makes CIFAR-scale dataset not suitable for showing the bad performance of SGD+RWP (validating theorem 7) and the advantage of AMA in the long term, which is the main purpose of this work. Hence, we argue that the experimented datasets in the main paper are more suitable for the research purpose of this work than CIFAR/MNIST, due to their large scale.
> >
> > ## We hope that the above responses resolve most if not all mis-understandings, and kindly ask the reviewer to reconsider the rating. And we are happy to have further discussions during rebuttal.

---

### Official Review · Reviewer_Ndan · 2022-10-24

**Confidence:** 3
**Correctness:** 3
**Technical Novelty And Significance:** 3
**Empirical Novelty And Significance:** 3
**Recommendation:** 6

**Clarity, Quality, Novelty And Reproducibility:**

Clarity: Excellent
Quality: Good
Novelty: Good
Reproducibility: Good

**Strength And Weaknesses:**

Strength:
This paper considers an interesting problem, it is clearly presented and readable. The results of numerous experimental results are convincing. The proposed method is easy to implement. A wide range of related studies are discussed.

Weaknesses:
1.The typesetting in some places is out of order, such as equations (23), (25), and (31).
2. In Page 4, "A4 bounds the degree of non-stationarity between consecutive iterations", why this assumption holds?
3. This author should add more description about the contribution of this paper.

**Summary Of The Paper:**

This paper studied the problem of improving information retention in online continual learning (OCL), the convergence of SGD in OCL are theoretical analyzed. Besides, an Adaptive Moving Average (AMA) Optimizer and a Moving-Average-based Learning Rate Schedule (MALR) to optimize the pure replay objective online are proposed. Experiment on several large scale continual learning benchmarks demonstrated the effectiveness of AMA+MALR.

**Summary Of The Review:**

This paper is well written and studies an interesting problem, I think it is marginally above the acceptance threshold.

---

> ### Author Response · Authors · 2022-11-11
> **We thank the reviewer for the detailed suggestion and the positive review. We will fix all minor raised minor issues in the revision and kindly ask the reviewer to consider raising the score if there is no major issue.**
>
> ## 1) In Page 4, "A4 bounds the degree of non-stationarity between consecutive iterations", why this assumption holds?
>
> We have discussed this in Appendix A.2 (the paragraph above all assumptions). We provide a more detailed explanation below:
>
> The purpose of Theorem 1 is to analyze the convergence speed of SGD given non-stationary objectives. It is obvious that if the problem allows the distribution to change arbitrarily between consecutive time steps, there is no way to design an effective optimization algorithm, since in the naïve example, we can always change all labels of the data in an adversarial fashion to make the loss always high for any algorithm.
>
> Hence, we introduce A4 in our assumptions to regularize the problem, so that it is possible to reduce the loss over time. What A4 says is that the loss difference between consecutive steps is limited. This is a reasonable assumption in practice, especially for OCL, since we usually train a model overtime on a gradually expanding data stream, and aim to perform well on all historical time steps. In this case, a small batch of new data arrives at each time step. Though the loss on this small batch might be high for some time steps, the amount of data in the current step is small comparing to all previously seen data, especially when the time step is large enough. In such cases, averaging the loss over all historical data makes $l_k$ not much different to $l_{k+1}$ especially when k is large enough.

---

### Official Review · Reviewer_axbU · 2022-10-25

**Confidence:** 3
**Correctness:** 3
**Technical Novelty And Significance:** 2
**Empirical Novelty And Significance:** 2
**Recommendation:** 3

**Clarity, Quality, Novelty And Reproducibility:**

The manuscript can be improved in terms of quality.
The proposed heuristics seem interesting but might not be enough novel to be published in this conference.

**Strength And Weaknesses:**

Indeed Online continual learning is a challenging scenario and the authors opted for an interesting research topic.

In my opinion this manuscript could be reorganized in a way that it is more to the point and explain how it is differentiated to the other relevant approaches in the past.
It would be interesting to discuss standard Adam and RMSProp and compare with the proposed heuristics in terms of time and performance.
Although it is mentioned that the authors used a population based search to find the best adaptive point within a convex hull for \gamma_k, I think making the effort to include the proof could indeed improve the work.
The overhead of the proposed approach seems to be high, how would you justify it in an online continual learning scenario.


**Summary Of The Paper:**


In Online continual Learning scenario, where one needs to deal with non-iid data stream, information retention can be a problem, even using unlimited replay buffer. In this study, applying naive Stochastic Gradient Descent and using constant or decreasing learning rate has been partly associated to the information loss in long term. The authors put forward using heuristics based on adaptive moving average to improve the optimization process, and also propose heuristics for learning rate scheduling.

**Summary Of The Review:**

In my opinion this study needs more work to be organized and to get into the shape to be published.

---

> ### Author Response · Authors · 2022-11-11
> **Thanks for the valuable comments. We believe most of them are already addressed in the paper, and refer to them below.**
>
> ## 1) In my opinion this manuscript could be reorganized in a way that it is more to the point and explain how it is differentiated to the other relevant approaches in the past.
>
> We really appreciate your comment, but what specific re-organization is good to make our manuscript much better? Can you provide any concrete suggestion and kindly indicate why that is much better than the way we present our work?
>
> In this work, we first propose theoretical analysis pointing out that in large scale OCL, plain SGD can be problematic even *without memory constraints*. This problem is not identified before and we propose AMA to solve it. We empirically verified the effectiveness of AMA in 3 large scale datasets CLOC (40M images), CGLM (580K images), ImageNet (1.2M images). We have compared our approach against both non-continual learning baselines (plain SGD, ADAM) and continual learning algorithms (GDumb, MIR and ER). We argue that we have clearly distinguished ourselves with existing literatures and demonstrated the contribution.
>
> ## 2) It would be interesting to discuss standard Adam and RMSProp and compare with the proposed heuristics in terms of time and performance.
>
> We emphasized that we have **already** discussed ADAM in the related work and compared against it in Appendix C.5. And we have even applied AMA on top of ADAM and observed also an improvement, though the SGD-based AMA performed the best. RMSProp is not frequently used in computer vision datasets, hence we did not compare against it. For runtime, AMA has almost the same runtime as SGD and ADAM since the MA model updates are parallel to the SGD updates in the implementation. We also included the overhead in terms of the number of operations in Sec. 4.3, as discussed later in response 4).
>
> ## 3) Although it is mentioned that the authors used a population-based search to find the best adaptive point within a convex hull for $\gamma_k$, I think making the effort to include the proof could indeed improve the work.
>
> As mentioned in the paper, we are not claiming that population-based search guarantees to find the best point in the convex hull, we propose it to search for **better MA weights than the initial constant**. Though a theoretical proof is left for future work, we have already included an empirical study in Fig. 3, showing that population-based search in AMA indeed can find better-than-initial MA weights during OCL.
>
> ## 4) The overhead of the proposed approach seems to be high, how would you justify it in an online continual learning scenario.
>
> We argue that with our careful algorithm design, the overhead of AMA is **not** high, and we have provided a complexity analysis in Sec. 4.3. Specifically, using the proposed AMA hyper-parameters, i.e., $K_M = 10$, $K_V = 20$, we only require per iteration 1/10 more forward pass iteration and 1/5 model weight update iteration (not including the gradient computation), which is small in practice. And these overheads are only in terms of the number of operations. In terms of runtime, the overhead can be hidden by parallelizing AMA updates with SGD updates, since they are independent and do not even need to be strictly synchronized. In the main experiments, we finished training 80 iterations per-batch on millions of images (40 million on CLOC and 1.2 million on ImageNet) for both SGD and AMA with similar time using our **un-optimized** experimental code (which would not be possible if AMA has strong overhead). To provide a more concrete idea on how fast AMA is, we will provide in Appendix C.8 of the revision the runtime of AMA compared to other methods, which we first summarize here. The runtime we show is the average wall-clock time for each method to finish 1 training iteration.
>
> AMA+MALR: 0.264s
> SGD+Cyclic: 0.245s
> SGD+RWP: 0.272s
> GDumb: 0.261s
> MIR: 0.395s
> ER: 0.327s
>
> Though we did not heavily optimize the implementation, and there are some noise in the runtime, the runtime of AMA is similar to that of SGD. This matches our complexity analysis and clearly shows the efficiency of AMA and supports our contribution on the algorithm acceleration (which seems to be even surprising according to your comment).
>
> ## In summary, we argue that both our findings in terms of 1) the problem of SGD in OCL with or without memory constraint, and 2) the proposed AMA algorithm to address this problem, are novel. And we kindly recommend the reviewer to reconsider his opinion and scoring.

---

### Official Review · Reviewer_mh9H · 2022-11-02

**Confidence:** 3
**Correctness:** 3
**Technical Novelty And Significance:** 2
**Empirical Novelty And Significance:** 2
**Recommendation:** 5

**Clarity, Quality, Novelty And Reproducibility:**

The paper is relatively clear, with good explainations of the reasoning behind the algorithm choices. The main improvements from a clarity perspective would be a better listing of all the tunable parameters and their effect, and more explainations on the gaps between the theoretical inspiration and the final algorithm.

The final algorithm is definetly novel, mostly as a new combination of existing techniques from optimization. The theoretical analysis is a straightforward extension of (Ghadimi & Lan, 2013) and (Besbes et al., 2015).

The paper is overall good quality, but falls slightly short since the final performance is not really compared with competitive baselines, and the theoretical analysis is quite far from the algorithmic part.

Despite the large numbers of hyperparameters, the authors make a good job in taking care of reporting them in the paper to help with reproducibility.

**Strength And Weaknesses:**

The strength of the papers are mainly in the simplicity of the resulting algorithm, and in the thorough exposition of the result (attempt at theoretically grounding the approach, multiple ablations, etc...)

The main weaknesses are:

a) the algorithm analysis is only for unlimited memory SGD. The final algorithm proposed is finite-memory and employs EMA. This is an enormous disconnect, and makes Thm. 1 more of a help to exposition in the paper than as a contribution. Section A.2 only gives a very brief and informal idea of how parts of this gap can be filled. The authors should either report a theoretical result more aligned with what the algorithm is actually doing, or clarify that Thm. 1 is mostly for illustrative purposes.

b) The final algorithm seems simple but hides its complexity behind several design choice. A partial list of values to tune includes K_W, K_V, K_M, delta, and how the train/validation splits should be chosen in practice both for AMA and MALR. The conditions C1-3 also introduce extra tunable parameters, without much discussion (e.g. what happens without C3?). Only a fraction of these values are ablated in the appendix. Giving advice on how to improve this is hard, as even the obvious thing (extensive ablation study) might change greatly from dataset to dataset.

c) The only baseline is SGD+RWP. This is enough to justify that the approach is superior to SGD+RWP, but many other approaches for OCL exist in the literature with varying degrees of complexity. Even a quick search yields several off-the-shelf tools to set up some baselines (e.g. https://github.com/ContinualAI/continual-learning-baselines) and at least a few of them should be included to gauge how much AMA-MALR can learn on these datasets.

**Summary Of The Paper:**

The paper introduces three main results:

- a theoretical analysis of SGD under distribution shift to highlight limitations of current OCL approaches
- a new OCL algorithm based on moving averages (AMA)
- an associated heuristic to tune the learning rate also based on moving averages (MALR)

other smaller contributions are a population-based approach to tune the EMA weights and empirical evaluation on multiple datasets

**Summary Of The Review:**

Overall the paper falls a bit short of the acceptance threshold due to the lack of focus in the contribution, where the authors try to provide results from three sides (theoretical, algorithmic and experimental) but only achieve small gains in each. In particular:
- The theoretical contribution is a straightforward extension of existing results, and only act as a guideline in designing the algorithm as it cannot be applied to it. This can be fixed by a more realistic analysis of AMA+MALR.
- The algorithmic contribution is novel and based only on familiar and well established tools, but its impact is reduced by the large number of hyperparameters involved in the actual implementation.
- The experimental results are good, but not rigorously evaluated against strong baselines and might be greatly influenced by the large number of hyperparameters.

---

> ### Author Response · Authors · 2022-11-11
> **Thanks for the valuable comments and confirmation of the novelty. We believe most of your concerns are already addressed in the appendix or the ImageNet experiment, which we explain below. We encourage you to take a further look at those parts of the paper and kindly reconsider your ratings.**
>
> ## 1. Theorem 1 for limited memory and its significance:
> Though we don’t limit the memory size during algorithm analysis, we explained theoretically how to interpret this result given limited memory in the last paragraph of Appendix A.2. You mentioned that ``the algorithm analysis is only for unlimited memory SGD. The final algorithm proposed is finite-memory and employs EMA. This is an enormous disconnect", this is not the case as we have tested our algorithm in **both** limited (Fig. 13) and unlimited memory (Fig. 1 and Fig. 5). Empirically, SGD+RWP behaves similarly as analyzed in Theorem 1 **with or without** memory limit. The unlimited memory case is already shown in Fig.1 and Fig. 5. For limited memory, we further provide the result of SGD+RWP for the ImageNet experiment, under the setup of Fig. 13. As shown in Fig. 13 of the revision, SGD+RWP performed badly in the long term due to the problem mentioned in Theorem 1, similarly to the case of unlimited memory. This result supports the applicability of our analysis.
>
> The novelty and significance of theorem 1 is that it reveals the problem of plain SGD on OCL **with or without** memory limit, as long as the computation is explicitly constrained. Existing CL algorithms are mostly designed to address the limited memory issue, which ignore the orthogonal problem shown in Theorem 1, and hence are not effective especially when the memory becomes less constrained (as shown in Fig. 5 and Fig. 13). This is the main motivation and novelty of this work, we are identifying a new problem that previous methods ignore, and propose new algorithms to address it.
>
> ## 2. Hyper-parameter robustness and the case without C3:
> Though several hyper-parameters are introduced in AMA+MALR, we use **the same setting for all these new hyper-parameters across 3 very different large scale datasets** (only tuned on CLOC), which shows the robustness, i.e., they **don't change greatly from datasets to datasets**. Most of these hyper-parameters ($K_W$, $K_V$, $K_M$) are used to reduce the computational overhead to make it comparable with SGD+RWP (see Sec. 4.3 for complexity analysis). The accuracy of the model remains stable if K_W, K_V and K_M are set smaller. $\delta$ remains effective between 2 to 10 in practice. $K_R$ in C1 and C2 is not introduced by MALR but the same as the one in RWP. We already discussed the detail on how to set $\sigma_k$ in practice. We argue that all these default values can be safely applied to new datasets, without (heavy) tuning.
>
> For the effect of each condition of MALR, we kindly note that they are **not** without much discussion. Sec. 5.2 has provided detailed discussion and analysis for the effect of each condition. E.g., C3 it is designed to address the problem P2 in Sec. 4.1, where we prevent the learning rate to be infinitesimal by monitoring the performance difference between MA and SGD model. We already showed in Fig. 2a that `` AMA+MALR+No C3 failed to limit the minimum learning rate (P2), and performed worse than AMA+MALR in the long term.”
>
> ## 3. The only baseline is SGD+RWP.
> We kindly argue that SGD+RWP is **not** the only baseline we have compared against, but we simply build AMA+MALR on top of it.
>
> For **non-continual learning** methods, we have compared AMA+MALR against 1) SGD+RWP (Fig. 1 & 5) 2) SGD+constant LR(SGD+CLR) (Fig. 1) 3) ADAM+RWP (Fig. 10) 4) SGD+Cyclic schedule (Fig. 5 & 13). Which includes different optimizers and different learning rate schedules.
> For **continual learning methods**, we have compared against the original CLOC algorithm [1] in the CLOC experiment (Appendix C.6.), and compared against GDumb [2], ER [3] and MIR [4] in the ImageNet experiment, for both limited (Fig. 13) and unlimited memory (Fig. 5). And we emphasize that it is clear from the ImageNet experiments that these continual learning methods are less effective than the most plain SGD when the limitation is less on the memory but more on the computation budget, since they are not designed to handle the problem of computation limitation. We did not compare GDumb, ER and MIR in CLOC and CGLM because their learning rate schedule is often task-based, i.e., they often set the same decaying schedule in each task, whereas CLOC and CGLM have a smooth stream without sharp task boundaries, making it unclear how to apply cyclic schedule on them.
>
> Though there are other continual learning algorithms at the time of our experiment, due to the implicitly increased computational cost, the competitors chosen above cover most algorithms that scales to datasets with millions of images.
>
> [1] Online continual learning with natural distribution shifts: An empirical study with visual data.
>
> [2] Gdumb: A simple approach that questions our progress in continual learning.
>
> [3] On tiny episodic memories in continual learning.
>
> [4] Online continual learning with maximally interfered retrieval.

---

### Decision · Program_Chairs · 2023-01-20

**Decision:**

Reject

**Justification For Why Not Higher Score:**

see above

**Justification For Why Not Lower Score:**

see above

**Metareview: Summary, Strengths And Weaknesses:**

Unfortunately, the reviewers were not enthusiastic enough about this paper for it to be considered for acceptance at ICLR 2023. There are just too many papers that reviewers were much more enthusiastic about this year, and ICLR has a very low acceptance rate. The authors are encouraged to take the reviewer comments very seriously, even if there are things you disagree with, and make sure that all issues are addressed, and any potential sources of confusion are completely eliminated, in the next version of the paper. Also, please ensure that these are addressed in the initial paper submission for that next conference.

**Summary Of Ac-Reviewer Meeting:**

see above